# Shallow landslides stability evaluation in loess areas according to Revised Infinite Slope Model: A case study of the 2013 "7.25" Tianshui sliding-flow landslide event in southwest of Loess Plateau, China

Jianqi Zhuang[1], Jianbing Peng[1], Chenhui Du[1], Yi Zhu[2], Jiaxu Kong[1],

[1]College of Geological Engineering and Geomatics/Key Laboratory of Western China Mineral Resources and Geological Engineering, Chang'an University, Xi'an, Shaanxi, 710054, China
[2]College of Land Engineering, Chang'an University, Xi'an, Shaanxi, 710054, China

*Correspondence to*: Jianqi Zhuang (jqzhuang@chd.edu.cn)

**Abstract.** The occurrence of shallow loess landslides induced by prolonged heavy rainfall is prevalent in loess-dominated regions, often leading to property damage, human casualties, and sediment pollution. Developing an accurate prediction model for shallow landslides in loess areas is crucial for effective landslide mitigation. In 2013, prolonged heavy rains from July 19th to the 25th triggered mass sliding-flow loess landslides in Tianshui, China. Landslide data, along with the characteristics of the sliding-flow loess landslides were obtained through extensive field investigations and remote sensing interpretations. The sliding-flow loess landslides event demonstrated clustering, high density, small areas, and long travel distance. The depth of the sliding surface is correlated with the saturated layer resulting from rainfall infiltration; typically less than 2 m deep and negatively correlated with slope steepness. Based on the common characteristics of shallow loess landslides, the mechanisms involved in the sliding flow landslide are proposed. The Revised Infinite Slope Model (RISM) was introduced using an equal differential unit method to address deficiencies when the safety factor remains constant or increases with increasing slope greater than 40° as calculated using the Taylor slope infinite model. The relationship between the critical depth and the slope of the shallow loess landslide was determined. The intensity-duration (*I-D*) prediction curve of the rainfall-induced shallow loess landslides for different slopes was constructed and combined with the characteristics of rainfall infiltration for use in forecasting regional shallow loess landslides. Additionally, the influence of loess strength on the shallow loess landslide stability has been analysed. The shallow loess landslide stability responds to slope and cohesion but is not sensitive to the internal friction angle.

## 1 Introduction

Loess is a porous and loose aeolian deposit of silt-sized particles mainly formed during the Quaternary period, and it is widely distributed in Asia, Europe, North America, and South America (Li et al., 2020). In China, loess is widely distributed, with an area of 630,000 km$^2$, accounting for 6.63% of the total land area in northwest, north, and northeast China. The loess deposit reaches a depth of up to 300 m, and the integrity and continuity of the loess layers are unparalleled globally (Liu, 1985).

However, environments dominated by loess are exceptionally fragile, experiencing significant soil erosion, topographical variation, concentrated rainfall, and have now emerged as one of the most developed geohazard areas in China (Derbyshire, 2001; Zhang and Liu, 2010; Zhuang et al., 2018). Loess is a distinctive soil characterized by its large pores, high compressibility, strong collapsibility, and high-water sensitivity, making it prone to surface failure (Xu et al., 2014; Li et al., 2007; Peng et al., 2015; Juang et al., 2018; Zhuang et al., 2022). The primary factor contributing to loess failure is the

interaction between water and loess, which can disrupt the loess structure and diminish its mechanical strength. In over 85% of loess landslides, water plays a crucial role, exhibiting characteristics such as early small-scale deformation, long run-out distance, unpredictable location distribution, rapid occurrence, and liquefaction. These factors lead to severe property damage and human casualties (Dijkstra, 1995; Wang et al., 2014; Zhuang et al., 2018; Zhu et al., 2022).

   Rainfall-induced slope failures are a common form of shallow landslides in the Chinese Loess Plateau (CLP). Field

investigations revealed that more than 50,000 landslides have occurred in the CLP in recent decades (Zhuang et al. 2018; Zhuang et al., 2022). Most of those landslides were triggered by prolonged heavy rainfall and with a slip surface depth of no more than 2 m (Zhuang et al., 2017; Zhuang et al., 2018; Zhuang et al., 2022). Precipitation infiltrates into the soil and percolates through the loess to a depth of approximately 2 m, attributed to the decreased infiltration rate with increasing depth (Tu et al., 2009; Xu et al., 2011; Zhuang et al., 2018). Shallow loess landslide events, due to prolonged heavy rainfall, mostly

occurred as sliding-flow landslides, occurring in Northern of the Shaanxi province in 2013 and 2017 and Tianshui of Gansu province in 2013 and 2015 (Peng et al., 2015; Wang et al., 2015; Zhuang et al., 2017; Zhang et al., 2020). Another impact of these landslides is the significant transportation of sediment into local rivers, resulting in river pollution, elevated riverbeds, and heightened flood risks.

   Numerous researchers have investigated rainfall infiltration, landslide mechanisms, and the prediction of rainfall-induced

shallow landslides. Among these studies, the most prevalent focus is on forecasting shallow loess sliding-flow landslides caused by rainfall, as this knowledge can be utilized for geohazard mitigation (Brenning, 2005; Bordoni et al., 2015; Ahmadi-adli et al., 2017; Reichenbach et al., 2018; Thomas et al., 2018; Cogan and Gratchev, 2019; Berti et al., 2020; Bordoni et al., 2020). Shallow landslide forecasting can be divided into three categories based on the prediction method. (1) Early warning of landslides through monitoring time-based deformation data (von Ruette et al., 2011; Galve et al., 2015; Roccati et al., 2018;

Segoni et al., 2018; Lombardo et al., 2020; Marino et al., 2020). The failure processes of most landslides advance through stages of deformation that gradually culminate in catastrophic failure. Throughout this progression, deformation is readily observable and monitorable, making it the most critical factor for landslide prediction. This approach is particularly pertinent for large-scale landslides with evident early deformation trends. (2) Forecasting landslides through temporal and spatial rainfall monitoring (Giannecchini, 2006; Salciarini et al., 2006; De Vita et al., 2012; Giannecchini et al., 2012; Cevasco et al., 2013;

von Ruette et al., 2013; Stähli et al., 2015). Rainfall is a key inducing factor of geohazards, and researchers have studied the critical rainfall values for intensity, duration, and total rainfall within specific areas for predicting landslides. (3) The utilization of statistical and qualitative methods to assess landslide susceptibility, in connection with the occurrence of landslides, can generate maps delineating hazard zones. These maps are valuable for land use planning and long-term forecasting (Jia et al.,

2008; Zizioli et al., 2013; Cevasco et al., 2014; Goetz et al., 2015; Guzzetti et al., 2006; Di Napoli et al., 2020; Di Napoli et

al., 2021; Keles and Nefeslioglu, 2021). However, it is important to note that statistical method results are largely dependent on the quality of data and the specific method employed. (4) Landslide early warning and forecasting based on physical modelling (Montgomery and Dietrich, 1994; Montrasio and Valentino, 2007; Formetta et al., 2016; Schilirò et al., 2016; Lizarrag et al., 2017; Wang et al., 2020; Leonarduzzi et al., 2021). Researchers have concentrated their efforts on elucidating the mechanisms and conditions that contribute to soil failure. The data for analysis is derived from soil tests, providing evidence

of the decay of cohesiveness and angles with precipitation infiltration (Skempton 1985; Iverson 2000; Baum et al. 2008; Baum and Godt 2010; Medina et al., 2021), as well as quantitative landslide assessment. Several physically based models have been proposed, including steady-state hydrology (SHALSTAB and SINMAP) (Montgomery and Dietrich, 1994; Pack et al. 1999), quasi-steady hydrology (dSLAM, IDSSM) (Dhakal and Sidle 2003), and Transient hydrology (TRIGRS) (Iverson, 2000; Baum et al. 2008).

However, due to the occurrence of shallow loess landslides on saturated or nearly saturated steep slopes that transition into loess flow, models such as TRIGRS, which is based on the infinite slope model and primarily designed for predicting shallow landslides in areas with gentle slopes, are challenging to apply in CLP (Wang et al., 2015; Zhuang et al., 2017). The SINMAP and SHALAD prediction models are also based on infinite slope models, with the sliding or soil depths being fixed parameters associated with landforms (Montgomery and Dietrich, 1994; Pack et al., 1999; Michel et al., 2020). According to

previous studies, shallow loess landslides with sliding-flow landslide characteristics are mainly induced by prolonged heavy rainfall and the sliding surface is the saturated or nearly saturated layer (Wang and Sassa, 2001; Zhang et al., 2013; Peng et al., 2015; Wang et al., 2015; Zhuang et al., 2017;Guo et al., 2019). Due to their small scale and difficult identification, shallow landslides on the CLP are a significant safety threat to local residential areas (Peng et al., 2015; Wang et al., 2015; Guo et al., 2019; Zhang et al., 2020). Given that the majority of shallow loess landslides occur within the upper 2 meters and transition

into high-speed, long-runout loess flows, the depth of the saturated layer plays a critical role in the study of loess sliding-flow landslides induced by prolonged heavy rainfall. However, there is currently no quantitative assessment for the critical slip depth, making it impossible to develop a rainfall criticality model based on physical processes.

The current study provides a comprehensive evaluation of the distinct characteristics of shallow loess sliding-flow landslides induced by prolonged precipitation. After adjusting for the constant or increasing safety factor with slopes greater

than 40°, calculated using the Taylor slope infinite model through the equal differential unit method, we propose the revised infinite slope (RISM) model. The objectives of this study were to determine the sliding depths of the saturated layer at different slopes, and, to consider rainfall intensity and duration for developing a shallow loess landslide prediction model using loess infiltration characteristics. The model was verified using the 2013 "7.25" Tianshui sliding-flow landslide events in Tianshui Gansu province, a sliding-flow loess landslide triggered by prolonged precipitation.

## 2 Study area

### 2.1 Geological and geomorphological

The study area is a hilly loess region located southwest of the CLP and is part of the transition zone between the Qinling Mountains and the Longshan Mountains (Peng et al., 2015; Zhang et al., 2020). The loess is deeply and widely distributed with vertical joints and fissures, creating a fragile geological environment where geohazards such as landslides and debris flows frequently occur (Peng et al., 2015; Zhang et al., 2020; Qi et al., 2021). The terrain of the study area is generally high in the southeast and low in the northwest, with altitudes ranging from 748 to 2,120 m and a relative height difference of 100 to 1,430 m. This geomorphic unit includes the loess hill area comprised of platforms (Yuan in Chinese), ridges (Liang in Chinese), domes (Mao in Chinese), and valleys. The loess hill area is dissected by the Wei River and Western Han Rivers along with their tributaries, featuring numerous ravines (Fig. 1). The hilltops of the loess hills form nearly horizontal surfaces at an elevation ranging from 1,900 to 2,000 m, with a relative height difference between the ridges of approximately 500 m. The Weihe River traverses the loess hilly area from west to east.

Tianshui region is located in the west wing of the Qilu-Helan Mountain zigzag structural system and the Qinling zonal structural belt. The sub-north-east-trending structural belts and Longxi structural insertions, folds, and compressive faults are highly developed in this area. The neotectonic movement is strong and the average annual elevation of the mountains is 0.6-8 mm. Additionally, the study area is located within the North-South and Tianshui-Lanzhou seismic belts, with frequent seismic activity (Sun et al., 2017).

### 2.2 Climate characteristics

The study area is located in a mid-latitude inland region and experiences a cold temperate semi-arid continental monsoon climate with four distinct seasons, characterized by dry winters and springs (low precipitation) and hot and humid summers (high precipitation). The annual average temperature is 10.6 ℃, while the annual precipitation ranges from 400 to 700 mm.

Elevation is the primary factor affecting the precipitation amount, with high precipitation in the high-altitude area and low precipitation in the low-altitude area. According to the precipitation data in recent decades years (Peng et al., 2015; Zhang et al., 2020), the greatest precipitation occurred in 1967 with 772.2 mm and the least precipitation occurred in 1939 with 316.6 mm. There are also fluctuations in the annual rainfall distribution. Most rainfall is observed from June to September, representing 70% of the annual precipitation (Peng et al., 2015; Zhang et al., 2020). In July, 2013, long-duration and large-scale heavy rainfall resulted in severe flooding and geological disasters.

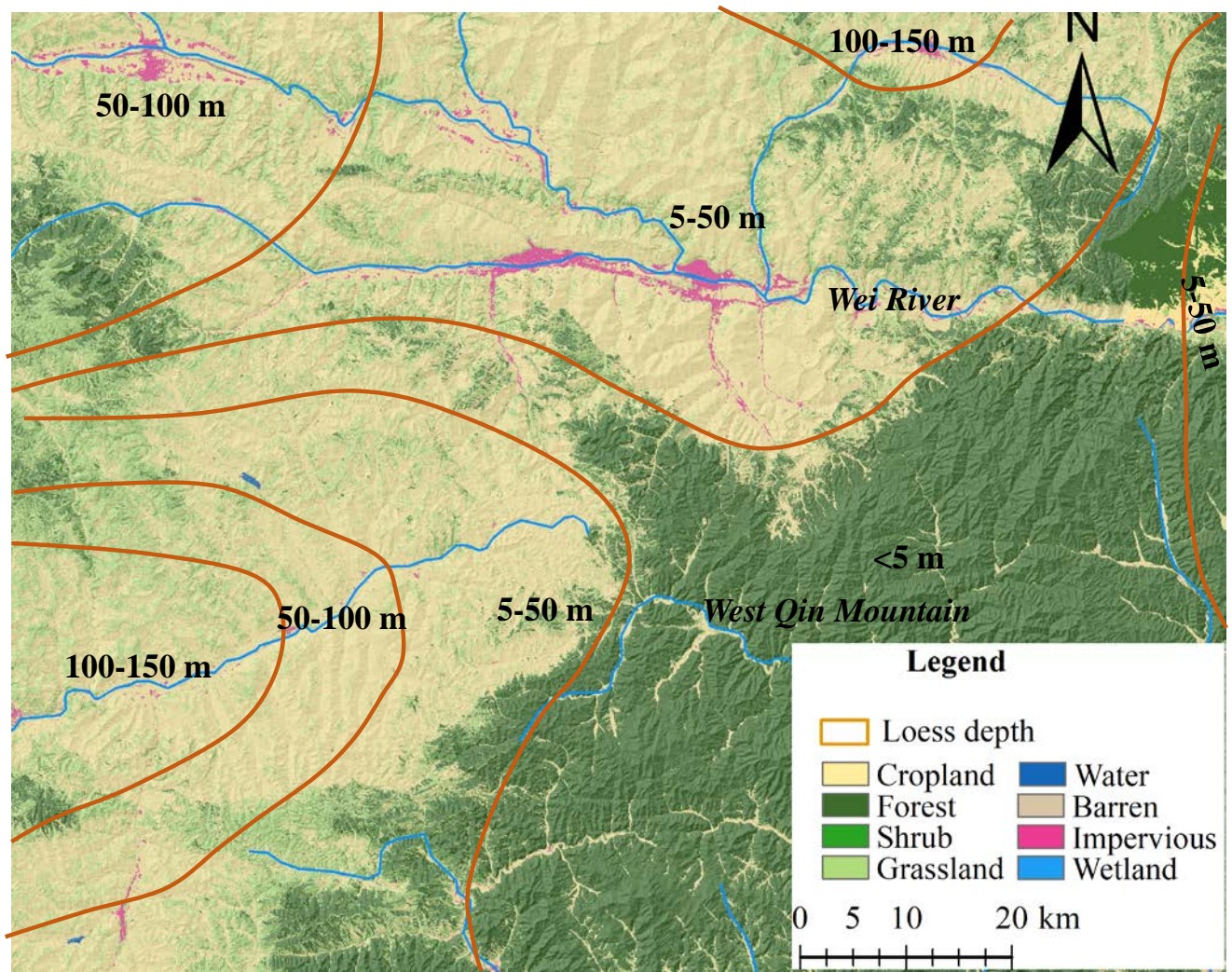

**Figure 1.** The geological and land use of the study area. Base DEM data from https://geocloud.cgs.gov.cn/#/home, land use data from http://www.geodata.cn.

Beginning on June 19, 2013, prolonged heavy precipitation occurred in the Tianshui region. The total cumulative precipitation and the maximum rainfall intensity of this event represent a 100-year return period (Peng et al., 2015; Qi et al., 2021). This heavy rainfall event lasted for 37 days and induced a mass sliding-flow landslide disaster. During this extreme rainfall event, there were four heavy rainfall stages, with different rainfall periods. The first heavy rainfall period began on June 19 at 19:00 and ended on June 21 at 4:00, with a cumulative rainfall of 285 mm, then from 3:00 to 20:00 on July 8 (128.9mm), followed by 16:00 on July 21 to 4:00 on July 22 (43.5mm), and the last period was from 23:00 on July 24 to 10:00 on July 25 (174.4mm) (Table 1). Over 45,000 shallow sliding-flow landslides were triggered, characterized by shallow and small scar areas, resulting in the death of 25 people.

**Table 1.** The precipitation data from the "7.25" Tianshui sliding-flow landslide events.

| Date | Accumulative rainfall / mm | Duration / h | Average intensity / mm/h | Max intensity/ mm/h |
|---|---|---|---|---|
| 19:00 on June 19 to 4:00 on June 21 | 285 | 34 | 8.382353 | 35.8 |
| 3:00 to 20:00 on July 8 | 128.9 | 17 | 7.582353 | 22.6 |
| 16:00 on July 21 to 4:00 on July 22 | 43.5 | 12 | 3.625 | 19 |
| 23:00 on July 24 to 10:00 on July 25 | 174.4 | 11 | 15.85455 | 32.2 |

**3 The "7.25" Tianshui sliding-flow landslide event characters**

**3.1 Landslide data**

High-precision remote sensing imaging data (~2 m resolution from Google earth images) from October 2012 (before the sliding-flow landslide event) and December 2013 (after the sliding-flow landslide event), along with field observations, was used to determine that a total of 47,005 sliding-flow landslides occurred in the study area. It can be seen from Figure 2 that the sliding-flow landslide distribution is primarily concentrated in the middle of the study area along the NNE direction and decreased gradually to the southeast and northwest.

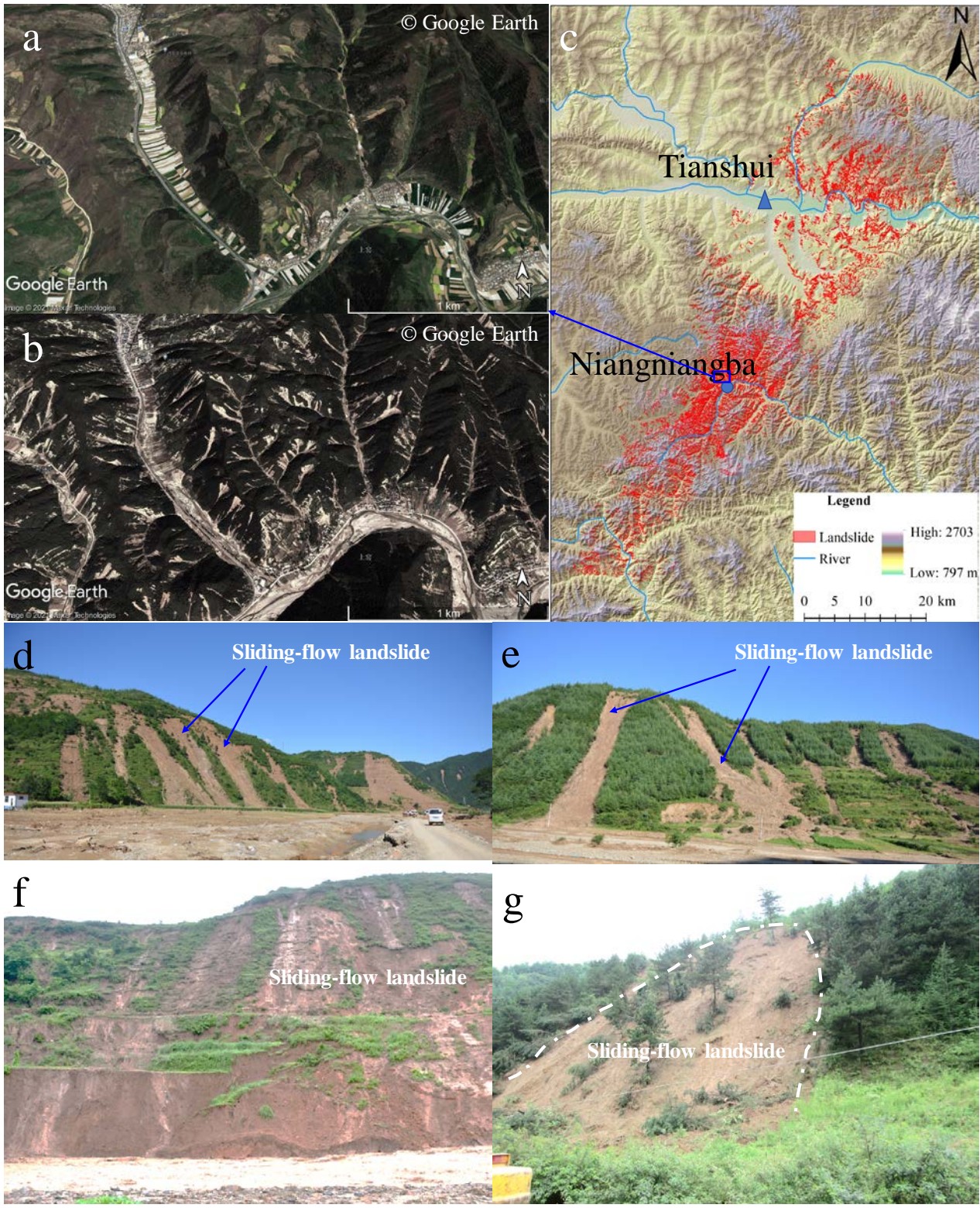

**Figure** 2 The landslides triggered by "7.25" Tianshui sliding-flow landslide events (a: before "7.25" Tianshui sliding-flow landslide events in October 2012 of Niangniangba town of Tianshui region; b: after "7.25" Tianshui sliding-flow landslide events in December 2013 of Niangniangba town of Tianshui region; c: the landslides distribution triggered by "7.25" Tianshui sliding-flow landslide events; d-g: typical sliding-flow landslides of "7.25" Tianshui sliding-flow landslide events. Base DEM data from https://geocloud.cgs.gov.cn/#/home)

The landslides occurred in the shallow loess layer at a depth of no more than 2 m and exhibited sliding-flow landslide characteristics as observed through extensive field investigations. Shallow landslides are typically triggered by prolonged heavy rainfall events, leading to a rapid increase in pore pressure or loss of cohesion (Iverson 2000; Wang and Sassa 2001; Sassa and Wang 2005; Tu et al., 2009; Zhang et al., 2013). Consequently, a failure surface develops within the soil profile or at the depth of precipitation infiltration. This indicates that the sliding soil layer is close to its liquid limit water content when the landslide occurs, resulting in flowing characteristics after slope failure.3.2 Size characters

The sliding-flow landslide impacted area is 65.69 km² and the density is greater than 25 sliding-flow landslides per/km². As shown in Fig. 2, the area of sliding-flow landslides in Niangniangba town of Tianshui region accounts for more than 35% of the total area, and shallow flow slips occurred on most of the slopes. The mean sliding-flow landslide area was 0.0013 km$^2$, which is smaller than what is typical for landslides triggered by rainfall or earthquakes (Table 2). Fig. 3 shows the proportion of landslides in different areas. It can be seen that most of the sliding-flow landslides were smaller than 2,000 m$^2$ and accounted for more than 80% of the total landslides. Landslides larger than 5,000 m$^2$ accounted for only 3% (Fig. 3), indicating that the "7.25" Tianshui sliding-flow landslide events were primarily small sliding-flow landslides occurring in groups.

**Table 2**. Landslide size and numbers triggered by earthquakes or rainfall in recent years.

| Items | Study area (km$^2$) | Number of landslide (km$^2$) | Landslide surface areas (km$^2$) | Average surface area of the landslide (per km$^2$) |
|---|---|---|---|---|
| Northridge earthquake (Harp and Jibson, 1996) | 10,000 | 11,111 | 23.8 | 0.00214 |
| Haiyuan Earthquake (Zhuang et al., 2018) | 40,000 | 3,700 | 117.45 | 0.03162 |
| Wenchuan Earthquake (Dai et al., 2011) | 41,750 | 56,000 | 811 | 0.01448 |
| Chi-Chi earthquake (Lin and Tung, 2004) | -- | 9,297 | 128 | 0.01374 |
| Kashmir earthquake (Owen et al., 2008) | 7,500 | 2,424 | -- | -- |

| | | | | |
|---|---|---|---|---|
| Umbria, Central Italy[rapid snowmelt] (Guzzetti et al., 2002) | 2,000 | 4,233 | 12.7 | 0.00301 |
| Guatemala [heavy rainfall] (Bucknam et al., 2001) | 10,000 | 9,594 | 29.5 | 0.00307 |
| "7.25" Tianshui sliding-flow landslide events | **1,936** | **47,005** | **65.69** | **0.0013** |

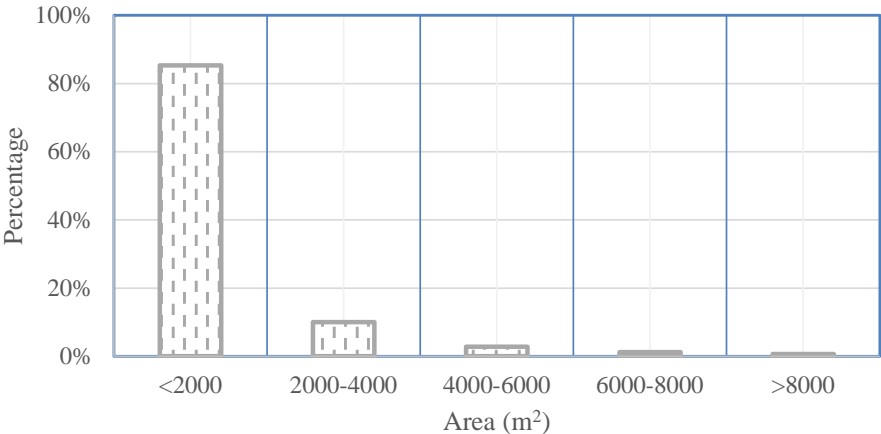

**Figure 3.** The proportion of "7.25" Tianshui sliding-flow landslides in different areas.

Fig. 4 depicts the "7.25" Tianshui sliding-flow landslide area, cumulative frequency distribution, and statistical significance of sliding-flow landslides with an area of failure $\geq$ 5,000 m$^2$, accounting for 90% of the total landslides and the largest landslide was only 0.099 km$^2$. We examined the area-frequency distribution of the "7.25" Tianshui sliding-flow landslide events via log-binning a normalized non-cumulative size-frequency distribution to plot frequency-density ($LgN = aA+ b$, where $N$ refers to the number of landslides in each bin) as a function of binned landslide area ($A$) (Fig.4) (Stark and Hovius, 2001; Malamud et al., 2004;). The higher value $a$ may reflect a greater ability to identify smaller landslides via high-quality imagery. The value for the "7.25" Tianshui sliding-flow landslide events ($a = -2.83$) is higher than the exponents reported for other coseismic inventories. For example, $a = -2.39$ for Northridge, California, $a = -2.30$ for Chi Chi, Taiwan, $a = -2.19$ for Wenchuan, China, and $a = -2.3$ for the average of event-based and historical inventories reported by Van den Eeckhaut et al. (2007) (Roback et al., 2018), also showing that landslides triggered by the "7.25" Tianshui sliding-flow landslide events were primarily small landslides occurring in groups.

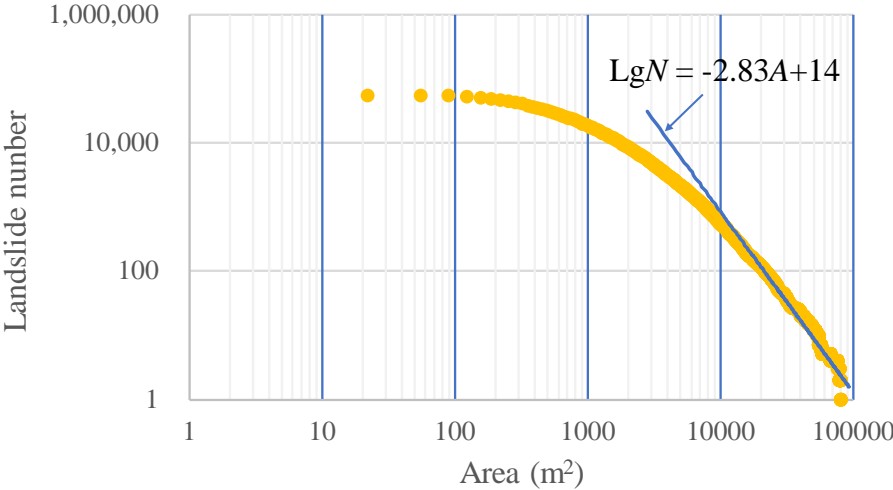

**Figure 4.** The cumulative frequency distribution of "7.25" Tianshui sliding-flow landslide area.

### 3.3 Mobility characteristics

We examined the probability densities of equivalent friction coefficients (landslide vertical height ($H$) / landslide travel distance ($L$)) for the "7.25" Tianshui sliding-flow landslide events using Matlab (Fig. 5). The landslide vertical height ($H$) was based on the altitudes of the highest and the lowest points, and the movement distance was the distance between the highest point and the lowest point. The above two parameters were obtained and calculated using ArcGIS spatial analysis.

The $H/L$ ratio frequency ratio of the "7.25" Tianshui sliding-flow landslide events, ranged from 0.01 to 0.88 with a mean of 0.32. According to a study by Wang (2000), landslide fluidization occurs when the equivalent friction coefficient is below 0.17. In our study, 16.85% of the loess sliding-flow landslides had equivalent friction coefficients below 0.17, indicating that the loess sliding-flow landslides moved with flow motion and resulted in longer sliding distances. For more than 96.27% of landslides, $H/L$ was less than 0.6 ($H/L < 0.6$ indicates a long-runout landslide) and belongs to long run-out landslides.

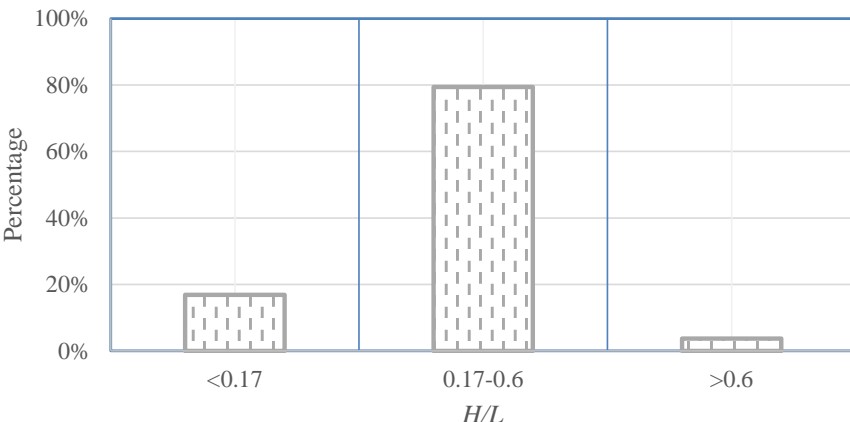

**Figure 5.** The probability densities of equivalent coefficients of friction of "7.25" Tianshui sliding-flow landslides.

To model the empirical relationship between sliding-flow landslide height and travel length, which was later verified

using images and field investigations, the aforementioned datasets were utilized to generate a plot of $H$ and $L$ on a single graph

(Fig. 6), where $L$ (x-axis) is the landslide travel distance and $H$ (y-axis) is the landslide height.

        The relationship between height difference and travel distance is positively correlated. With an increase in the height

differential, the travel distance increased and the slope of the fitting trend line between the height difference and travel distance

was 0.37, indicating that the travel distance is greater than the height difference and shows evident long-runout travel distance

characteristics.

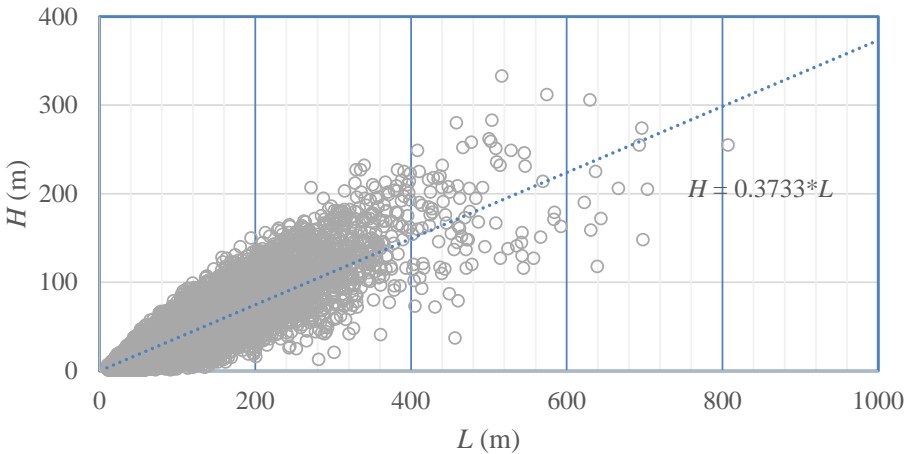

**Figure 6.** The relationship between height differential and travel distance of the "7.25" Tianshui sliding-flow landslides.

### 3.4 Sliding depth characteristics

To obtain sliding-flow landslide depths, we conducted a detailed investigation of sites throughout the study area, such as

Niangniangba town of Tianshui region, and obtained depth data for 83 landslides based on the characteristics of the sliding-

flow landslide and the thickness of the loess scar at the edge of the landslide. Fig. 7 displays the distribution of sliding-flow

landslide depths and area; no landslide had a depth greater than 2.0 m and over 70% ranged from 0 to1 m (Fig. 7). Additionally,

we found that the depth of the sliding-flow landslide had no correlation with the landslide area and had a negative correlation

with the slope. With increasing slope, the depth of the landslide decreases, that is, the greater the slope, the shallower the

sliding surface, and the smaller the slope, the greater the depth of the sliding surface.

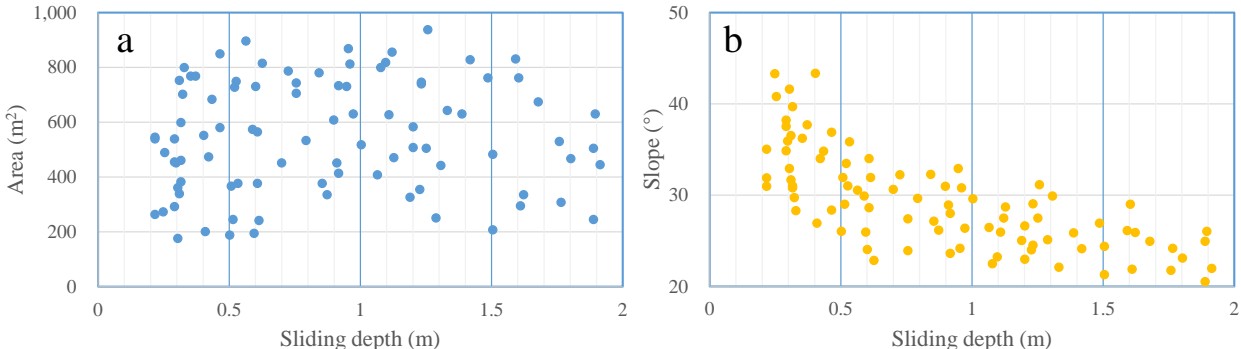

**Figure 7.** The distribution of sliding-flow landslide depths and area of the "7.25" Tianshui sliding-flow landslides, a) landslide surface area vs sliding surface depth and b) slope vs sliding surface depth.

Meanwhile, the distribution of slope and sliding-flows landslides in the study area was analyzed statistically. It was observed that the majority of sliding-flows landslides had a gradient value within the range of 30–50°, while landscape areas were concentrated in the range of 10–30°. Furthermore, sliding-flows landslides with slopes greater than 50° accounted for 13.81% in the study area. Consequently, there is a higher probability of failure for sliding-flow landslides on steep slopes (50°) in the CLP.

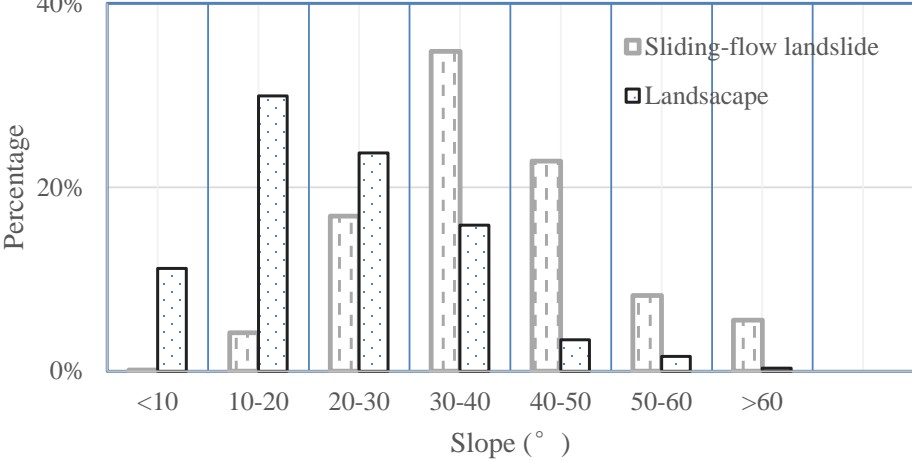

**Figure 8.** Map showing the relationship between the sliding-flows landslide, landscape and slope.3.5 The shallow sliding-flow landslide formation process

Force analysis shows that a slope will fail when the gravity component along the slope direction is greater than the shear strength of the soil (Sassa, 2000; Ochiai et al., 2004; Gabet and Mudd, 2006). If the stability coefficient of the slope is high and the soil reaches its liquid limit water content before the failure, the slope will spontaneously liquefy and flow during the sliding process (Wang and Sassa, 2001; Wang et al., 2015). Usually, the ratio of pore-water pressure to the total normal stress

of the soil is used to express the soil liquefaction ratio. When the liquefaction rate is 1, the pore-water pressure equals the normal stress of the soil and the failing slope is in a state of complete liquefaction (Hungr et al., 2001; Wang and Sassa, 2001; Sassa and Wang, 2005; Wang et al., 2015). Typically, the normal stress of shallow landslides is low and the slope deformation is attributed to decreased strength. The soil has an obvious volume reduction due to the large pore structure of the collapsing soil structure, followed by a sharp increase in the pore-water pressure (Wang and Sassa, 2001; Sassa and Wang, 2005; Peng et al., 2018). Shallow sliding-flow landslides are triggered by prolonged heavy precipitation and rainfall infiltration to a certain loess depth (mostly 1-2m) close to liquid limit water content, followed by a decrease in the soil strength (Tu et al. 2009; Xu et al., 2011; Zhuang et al. 2018). When the anti-slip force of the saturated soil is less than the sliding force, the saturated soil layer will fail. Due to the large pore structure of the soil, once soil deformation occurs, the water in the pores cannot be released, resulting in the pore-water pressure increasing sharply which causes liquefaction of the saturated soil to form a mudflow (Fig. 9).

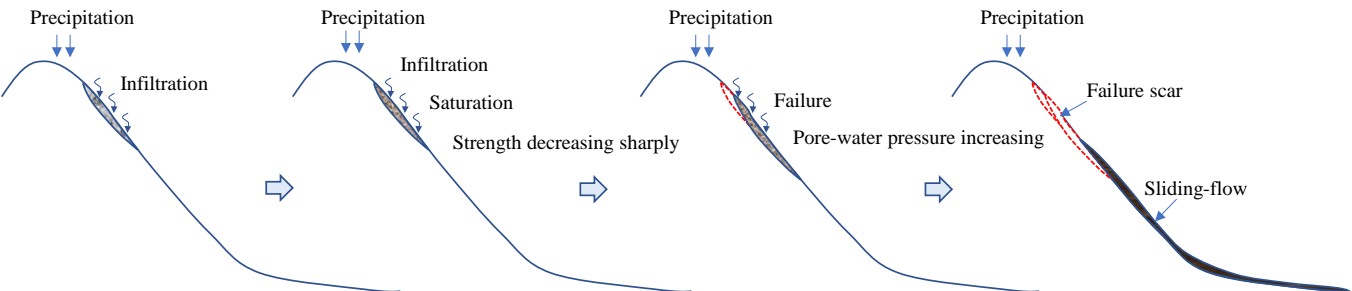

**Figure 9.** The shallow sliding-flow landslide formation process.

## 4 Forecast model of shallow loess landslide development

### 4.1 Infinite Slope Model

According to field investigations and other research results, the shallow sliding-flow landslides are concentrated within 2 m of the surface and are negatively correlated with slope (Tu et al. 2009; Xu et al., 2011; Zhuang et al., 2017; Zhuang et al. 2018; Zhuang et al., 2022). Therefore, infinite slope models that take into account the soil layer thickness and maximum infiltration depth cannot be directly applied in shallow landslide assessments. Previous studies have indicated that short-duration heavy precipitation has a lesser impact on the stability of loess slopes, while prolonged heavy precipitation significantly increases water content and reduces slope stability (Wang and Sassa, 2001; Wang et al., 2015; Peng et al., 2018). Shallow landslides transformation to mudflows occurs most often on slopes of 25 to 45°, and the sliding body is close to the liquid limit water content before slope instability. Pore-water pressure is a key factor for soil slope failure (Iverson 2000). Previous studies have demonstrated that pore water pressure is the primary cause of soil landslides, and excess pore-water pressure serves as the triggering factor for fluidization (Iverson et al., 1997; Sassa and Wang 2005; Gabet and Mudd, 2006). The failure of the sliding

body is mainly attributed to a reduction in soil strength, leading to an increase in the sliding force exceeding cohesive forces following rainfall infiltration that saturates the soil (Wang et al., 2015; Peng et al., 2018).

According to the infinite slope model (Fig. 10), Taylor proposed the following safety factor equation (Taylor, 1948):

$$K = \frac{(\gamma_{sat}-\gamma_w)h_w \cos^2 \alpha \tan \phi' + c'}{\gamma_{sat}h_w \sin \alpha \cos \alpha},$$    (1)

where $a$ is the slope angle, $\gamma_w$ is the soil floating weight, $\gamma_{sat}$ is the saturation weight of the soil, $b$ is the length of the sliding body, $h_w$ is the depth of the sliding surface, and the approximate depth of loess at the liquid limit water content due to infiltration, $c'$ (cohesive), and $\phi'$ (internal friction angle) is the effective strength index of the soil at the sliding surface.

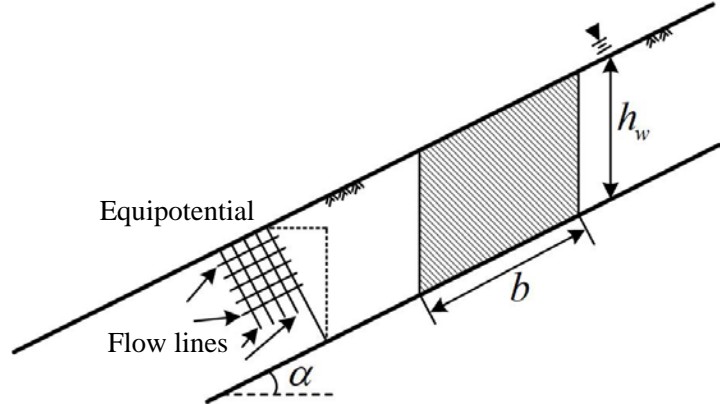

**Figure 10** The cross-section of an infinite slope (Modified from Taylor, 1948).

Using the mean strength of the saturated loess, the stability factor is calculated according to the infinite slope model proposed by Taylor (Taylor, 1948). The results are shown in Fig. 11.

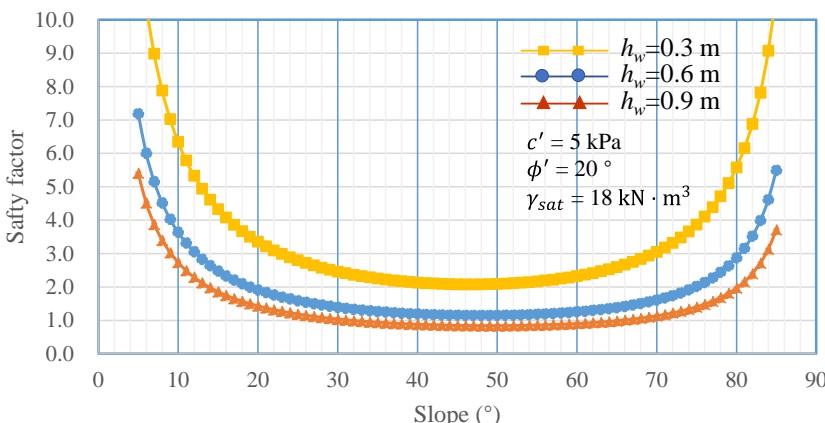

**Figure 11.** The safety factor according to the infinite slope model proposed by Taylor (Taylor, 1948).

Figure 11 shows that for a cohesive soil slope, the safety factor decreases with increasing slope when the slope is less than 40°. There is little difference for slope from 40 to 50°; however, the safety factor increases when the slope is great than 50°. The calculated results exhibit discrepancies with the actual conditions, particularly in the aspect that the

model indicates an increase in safety factor with increasing slope. However, most shallow landslide physical prediction

models (e.g., TRIGRS, SINMAP, SHALAD) are based on Taylor's infinite slope model. As a result, these models cannot be effectively applied to areas with high slopes (>40°), limiting their widespread applicability (Montgomery and Dietrich, 1994; Baum et al., 2008; Zhuang et al., 2017).

## 4.2 Revised Infinite Slope Model

To ensure the consistency of Taylor's infinite slope model calculation results with actual conditions, this study has made

modifications to the model using an equal differential unit method. As depicted in Fig. 12, by maintaining a constant depth of the saturation zone as the slope increases, the formula for calculating the self-weight of the soil strip unit has been revised as follows::

$$W = \gamma_{sat} h_w (b + \Delta b) \cos(\alpha + \Delta\alpha), \tag{2}$$

Where $b$ is the length of the sliding body and $\Delta\alpha$ is the value of the increasing slope.

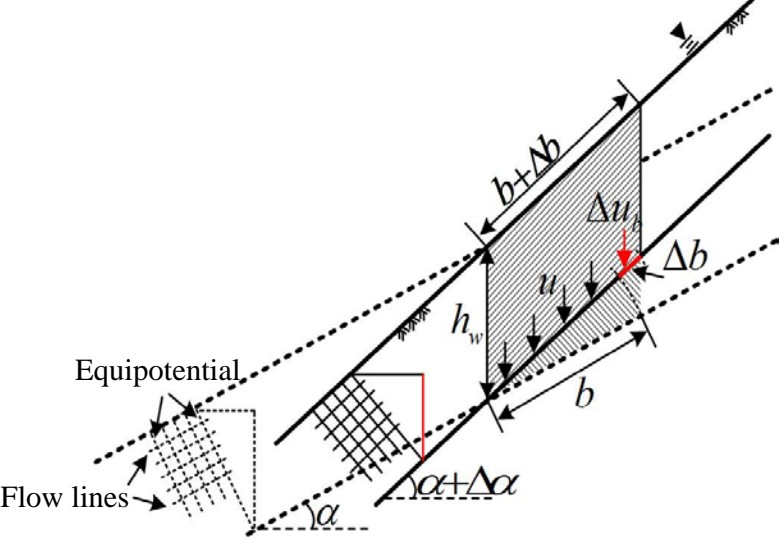

**Figure 12.** Figures demonstrating the infinite slope model with variation in slope.

In Eq. (2), the self-weight of the soil strip unit and the control condition of the soil strip area are unchanged:

$$h_w(b + \Delta b) \cos(\alpha + \Delta\alpha) = h_w b \cos\alpha, \tag{3}$$

Eq. (2) shows that, although the change of slope does not affect the self-weight of the differential element, it leads to a

change in the bottom area of the element, which causes the cohesion strength and pore water pressure of the bottom surface of the soil strip element to change. The change in slope only influences the component values of forces in normal and tangential directions on the sliding surface. Therefore, when evaluating slope stability with different slopes, it is essential to account for changes in cohesion strength and pore water pressure caused by an increase in slope $\Delta\alpha$.

The change in cohesive strength ($\Delta C$) and pore water pressure ($\Delta u_b$) resulting from increasing the slope of $\Delta \alpha$ are described as:

$$\Delta C = c' \Delta b, \tag{4}$$

$$\Delta u_b = \gamma_w h_w \Delta b \cos^2(\alpha + \Delta \alpha), \tag{5}$$

The effective normal stress ($N'$) and effective anti-sliding force ($T_f$) at the bottom of the soil strip element due to the slope increasing from $\alpha$ to $\alpha + \Delta \alpha$ is expressed as:

$$N' = \gamma_{sat} h_w (b + \Delta b) \cos^2(\alpha + \Delta \alpha) - \gamma_w h_w (b + \Delta b) \cos^2(\alpha + \Delta \alpha), \tag{6}$$

$$T_f = [\gamma_{sat} h_w (b + \Delta b) \cos^2(\alpha + \Delta \alpha) - \gamma_w h_w (b + \Delta b) \cos^2(\alpha + \Delta \alpha)] \tan \phi' + c'(b + \Delta b), \tag{7}$$

The sliding force of the soil strip unit is changed to:

$$S = \gamma_{sat} h_w (b + \Delta b) \cos(\alpha + \Delta \alpha) \sin(\alpha + \Delta \alpha), \tag{8}$$

The revised effective anti-slip force according to the Eqs. (4), (5), and (7) is:

$$T_f = [\gamma_{sat} h_w (b + \Delta b) \cos^2(\alpha + \Delta \alpha) - \gamma_w h_w b \cos^2(\alpha + \Delta \alpha)] \tan \phi' + c'b, \tag{9}$$

Combining Eqs. (8) and (9), the safety factor can be obtained:

$$K = \frac{[\gamma_{sat} \cos \alpha - \gamma_w \cos(\alpha + \Delta \alpha)] h_w \cos(\alpha + \Delta \alpha) \tan \phi' + c'}{\gamma_{sat} h_w \cos \alpha \sin(\alpha + \Delta \alpha)}, \tag{10}$$

Making $\alpha = \alpha_1$, $\alpha + \Delta \alpha = \alpha_2$, $(\alpha_1, \alpha_2) \in \alpha$, Eq. 10 can be expressed as:

$$K = \frac{(\gamma_{sat} - \gamma_w m_\alpha) h_w \cos \alpha_2 \tan \phi' + c' \sec \alpha_1}{\gamma_{sat} h_w \sin \alpha_2}, \tag{11}$$

Making $m_\alpha = \frac{\cos \alpha_2}{\cos \alpha_1}$, for the simple equation, the term $c' \sec \alpha_1$ in Eq. (11) can be defined as the reference cohesion strength. Since the value of $c' \sec \alpha_1$ varies monotonically with $\alpha_1$ changing, to make the reference cohesion strength corresponding to any slope angle equal, it only needs to satisfy that $\alpha_1$ is less than or equal to $\alpha_2$:

$$\alpha_1 = min\{\alpha_2\}, \alpha_2 \in \alpha, \tag{12}$$

Therefore, the safety factor of the RISM can be expressed as:

$$K = \frac{(\gamma_{sat} - \gamma_w \cos \alpha) h_w \cos \alpha \tan \phi' + c'}{\gamma_{sat} h_w \sin \alpha}, \tag{12}$$

Using the mean strength of saturated loess in the loess area, the stability is calculated according to the RISM proposed by this study, and the results are shown in Fig. 13. According to the using equal differential unit method, the RISM corrects the safety factor that increases with the slope increasing when the slope is larger than 50° (calculated using the Taylor slope infinite model). The results obtained by the RISM maintain consistency with the calculation result of the Taylor method at low angles, and the calculated results decrease with an increased slope when the slope is larger than 40°.

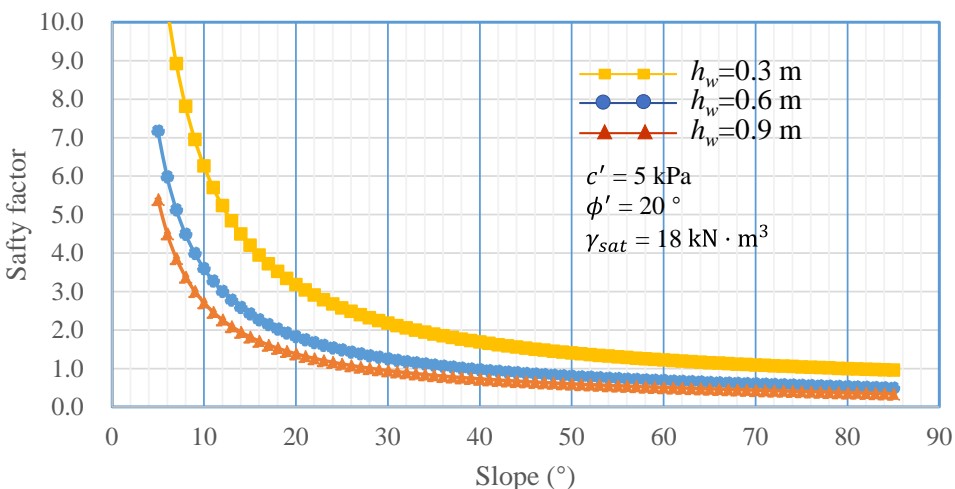

**Figure 13.** The safety factor according to the RISM.

## 4.3 Critical depth of shallow loess landslides

When the stability coefficient $K$ is 1 in Eq. (12), the critical depth of loess close to liquid limit water content, or the sliding surface of the shallow loess landslide, can be obtained:

$$h_{cr} = \frac{c'}{\gamma_{sat} \sin \alpha - (\gamma_{sat} - \gamma_w \cos \alpha) \cos \alpha \tan \phi'}, \qquad (13)$$

The critical liquid limit water content depth, or the sliding surface of the shallow loess landslide, for different slopes can be determined by obtaining the soil strength of the nearly saturated state. Upon reaching the critical depth of liquid limit water content, the layer will experience failure. The cohesion and angle of internal friction were tested using the triaxial test method at saturation following the soil test standard GBT50123-1999. According to the loess test results, the cohesive forces of undisturbed loess at the liquid limit water content (0-2 m) range from 3 to 9 kPa with an average of 5 kPa, and the internal friction angle ranges from 11 to 21° with an average of 15° (Fig. 14). Therefore, the relationship between the critical approximate liquid limit water content depth and the slope can be calculated. With increasing slope, the critical approximate liquid limit water content layer gradually decreases, but the rate of decrease slows, from 1.14 m at 20° to 0.47 m at 40° (Fig. 14). This relationship can be expressed via the power-law as:

$$h_w = 34.13a^{-1.15}, \qquad (14)$$

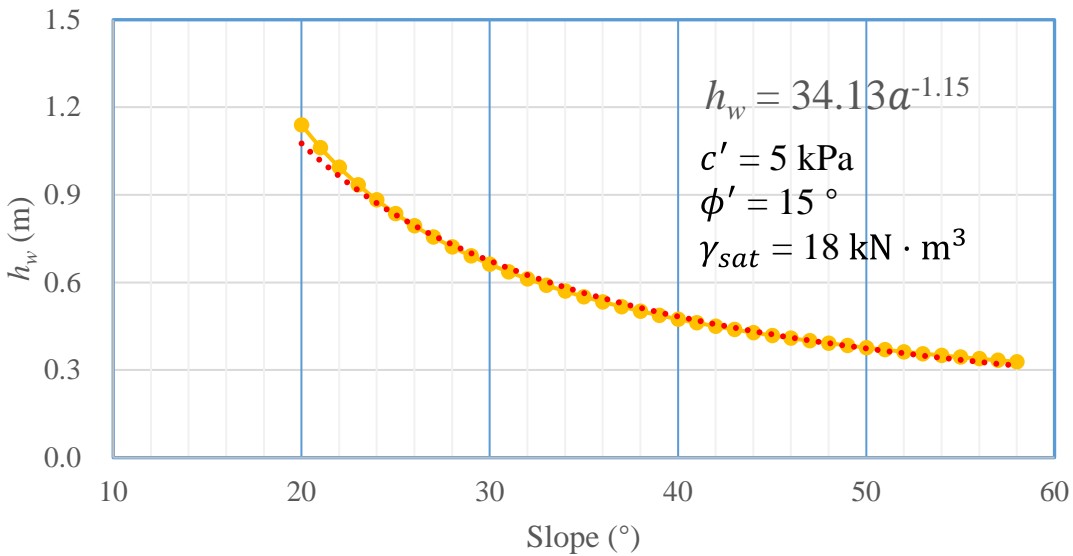

**Figure 14.** The relationship between the critical depth and the slope.

### 4.4 The *I-D* curve of the loess shallow landslide

The critical rainfall intensity-duration (*I-D*) method is often used in forecasting rainfall-induced shallow landslides (Guzzetti et al., 2007; Guzzetti et al., 2008; Baum and Godt, 2010; Zhuang et al., 2015). Thresholds empirically derived from rainfall intensity-duration have been widely used to identify rainfall conditions that result in the occurrence of landslides (Guzzetti et al. 2007; Guzzetti et al. 2008; Baum and Godt 2010). Inspection of the *I-D* thresholds reveals the general form:

$$I = \beta D^b + c,  \tag{6}$$

Where *I* is the mean rainfall intensity, *D* is rainfall duration, and $c$, $\beta$, and $b$ are other parameters. For the majority of *I-D* thresholds, $c = 0$, and Eq. 6 takes the form of a simple power law.

$$I = \beta D^b,  \tag{7}$$

Previous studies have created statistical *I-D* curves based on landslide data and rainfall data. However, these empirical models require many years of precipitation data and calibration parameters. Even so, the determination of the landslide

threshold *I-D* curve is inaccurate due to the uncertainty of rainfall monitoring, such as the location, quantity of monitoring sites, and the definition of the start and end time of rainfall events (Zhuang et al., 2015; Guo et al., 2016).

The critical *I-D* curve for slope instability for different slopes can be constructed using the critical approximate liquid limit water content depth model combined with the saturated infiltration characteristics of loess in the study area. Six infiltration tests were carried out in the study area using the single-ring infiltration test to determine the infiltration coefficient

(*i*) of loess under rainfall. The constant infiltration rates were: 38.6, 33.1, 39.1, 32.2, 36.2, and 31.6 mm/h. Additionally, it was

observed that the time between initial infiltration and stable infiltration is less than 10 min. Therefore, in this study, the average stable infiltration rate of 36.0 mm/h was selected as the infiltration coefficient.

$D= h_w /i,$    for the precipitation intensity is higher than the infiltration coefficient (8)

$D=h_w /I,$  for the precipitation intensity is lower than the infiltration coefficient (9)

To model the relationship between $I$ and $D$, the two variables were plotted on a single graph, where $D$ (x-axis) is the rainfall duration and $I$ (y-axis) is the rainfall intensity. According to the critical sliding surface depth under different slopes (Eq. 14), curves $I$ and $D$ under different slopes can be calculated by using eq. (7), (8) and (9). The $I-D$ curve of the different slopes in the area can be obtained based on the infiltration coefficient and the sliding depth (Fig. 15).

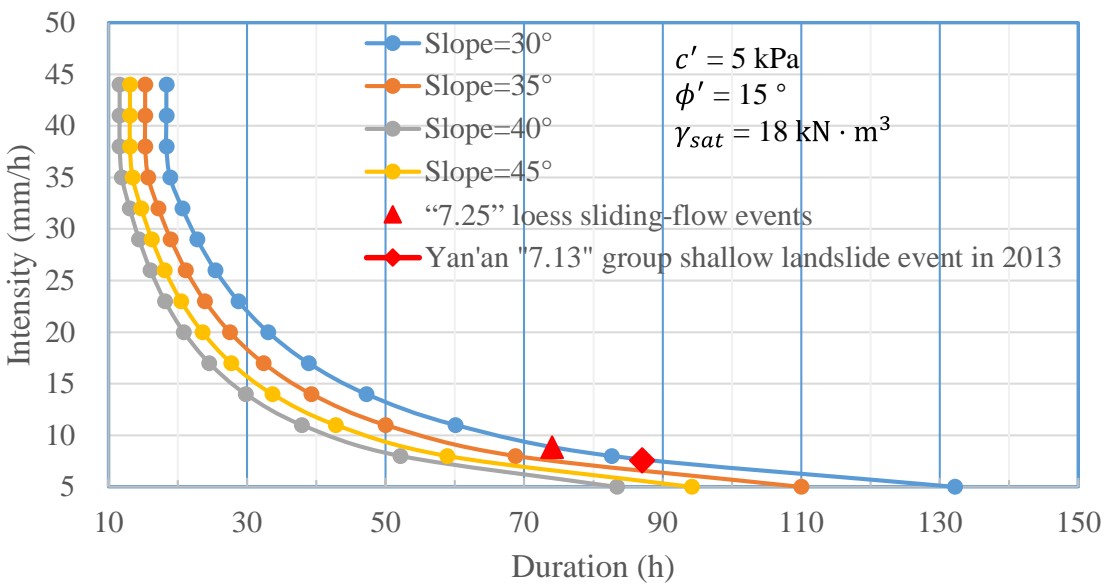


**Figure 15.** The $I-D$ curve of the different slopes in the loess area.

**5 Discussion**

**5.1 Model compare**

Upon comparing the $I-D$ curves for different slopes with existing models, it was observed that the $I-D$ curves derived from the
physical model in this study exhibit higher values compared to other statistical and probabilistic models. Additionally, while the $I-D$ curves for other regions are obtained through statistical methods, any individual landslide occurrence within this area is considered as a separate event. Meanwhile, many researchers have pointed out that antecedent rainfall plays a significant role in triggering landslides in loess areas which is different for other regions, such as Hong Kong, fire impacted areas in the

US, and the southwest mountains in China (Cui et al., 2008; Zhuang et al., 2015). Therefore, the *I-D* curves of other areas will
be lower than the *I-D* curve constructed based on physical models (Fig. 16).

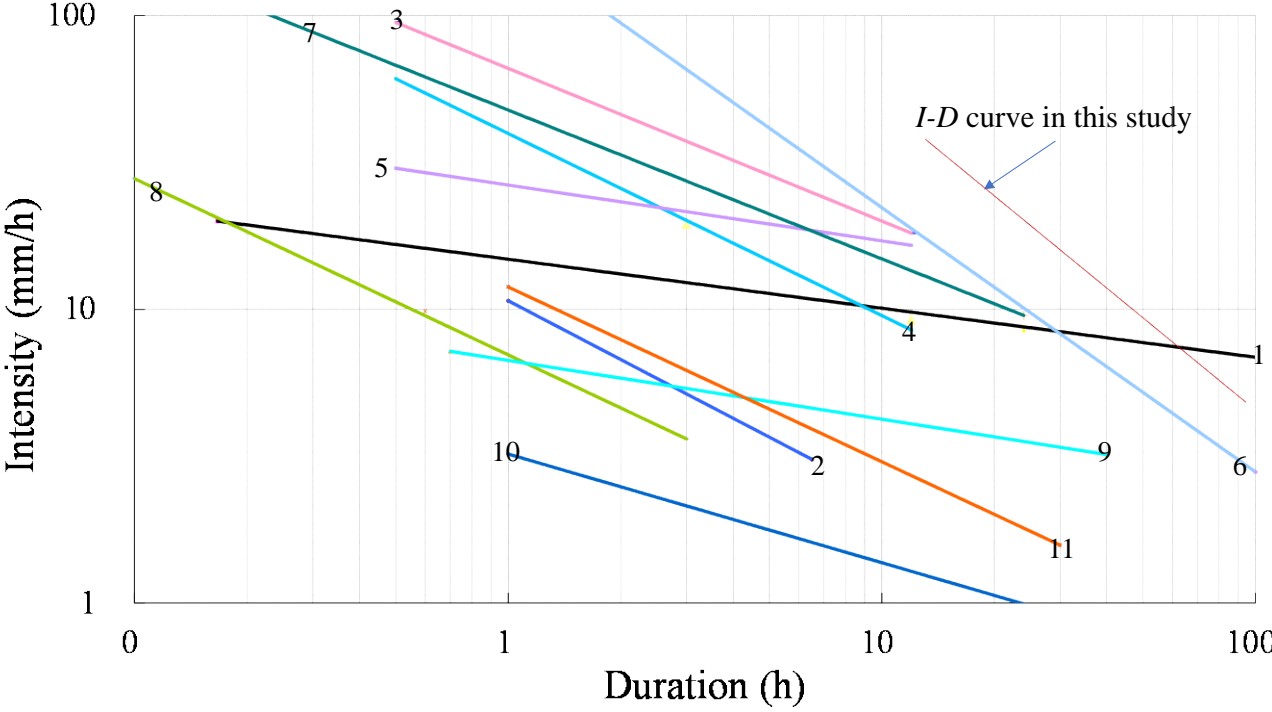

**Figure 16.** Comparison of I-D curves from the current and prior studies (No.1: Caine 1980; No. 2: Wieczorek 1987; No. 3, 4, 5: Jibson 1989; No. 6: Guadagno 1991; No. 7: Paronuzzi et al. 1998; No. 8: Crosta and Frattini 2001; No. 9: Shieh et al. 2009; No. 10: Guo et al. 2013; No. 11: Zhuang et al., 2015).

From the distribution of shallow landslides in this area, it can be seen that the shallow landslides in this area mainly occur on slopes of 35 to 50°. Marking the rainfall duration and intensity of the "7.25" Tianshui sliding-flow landslide events and the "7.13" group shallow landslide in Yan'an in 2013 (Wang et al., 2015; Zhuang et al., 2017) on the *I-D* line, it can be seen that the rainfall duration and intensity of both events are above the *I-D* curve of 35° (Fig. 15), indicating that the constructed curve is reliable and can be used to forecast shallow landslides in this area.

**5.2 Model limitations**

         Previous studies have assumed that restoration of vegetation is beneficial for preventing landslides, mainly because of the reinforcement effect of roots on soil (Waldron, 1977; Roering et al., 2003). However, the inhibitory effect of vegetation on landslides is gradually being questioned due to research on the negative relationship between vegetation and landslides in recent mass landslides in loess and other well-vegetated areas. For example, the "9.16" mass shallow landslides in Changning, 380  Yunnan in 2014, the Yan'an "7.13" mass shallow landslides event, the 2015 Tianshui mass shallow landslides event, and the

2010 Nanping mass shallow landslides event in Fujian Province all had mass shallow landslides induced by prolonged heavy precipitation, often occurring in areas well-vegetated areas (Fig. 17, Wang et al., 2015; Peng et al., 2015; Zhuang et al., 2017; Zhang et al., 2020). Due to the complexity of the influence of vegetation on slope stability, most of the studies have not clarified whether the vegetation induces or inhibits landslides (Rickli and Graf, 2009; Preti, 2013; Zhuang et al., 2022). In general, plant roots can reinforce the slope soil and improve the shear strength of the soil (Waldron, 1977; Roering et al., 2003). However, the increase of vegetation can also result in greater stored precipitation and infiltration into the soil (Wang et al., 2015; Zhuang et al., 2017; Zhuang et al., 2022). Plant root systems also provide channels for preferential flow, resulting in greater soil water contents which promote the occurrence of shallow landslides (Zhuang et al., 2017; Zhuang et al., 2022). The vegetation on the CLP is often shrubs with shallow root systems that do not exceed the sliding surface. Since most shallow landslides slide along the bottom of the root system, according to field investigations, they do not prevent these landslides (Wang et al., 2015). Slopes with good vegetation coverage increase the weight of the sliding body during precipitation. The prediction model proposed in this study does not consider the impact of vegetation on shallow landslides. According to data from monitoring numerous loess areas, increased restoration of vegetation results in more rainwater infiltrating into the loess, increasing the water content of the shallow loess, and results in an increased possibility of shallow landslides (Xu et al., 2024). There is not only downward vertical infiltration but also horizontal infiltration along the slope direction in the process of rainfall infiltration on the slope surface resulting in the confluence of water in the slope body (Wang et al., 2015; Zhuang et al., 2022).

At the same time, according to the assumption of the infinite slope model, it is mainly applicable to the slope extent is much larger than the depth of potential slip surface, slope angle is not too large, and soil characteristics are homogeneous along the slope direction, which means that the infinite slope model is only suitable for homogeneous soil slope instability prediction due to rainfall-induced (Taylor, 1948). For slopes constituted of binary structures slope and failure controlled by the strength of the structure of the rock-soil mass is often not able to evaluate a correct factor of safety for the slope. So, the RISM is also suitable for homogeneous soil slopes, and slope extent is undefined or much larger than the depth of potential slip surface, but can be applied to steeper slopes in CLP.

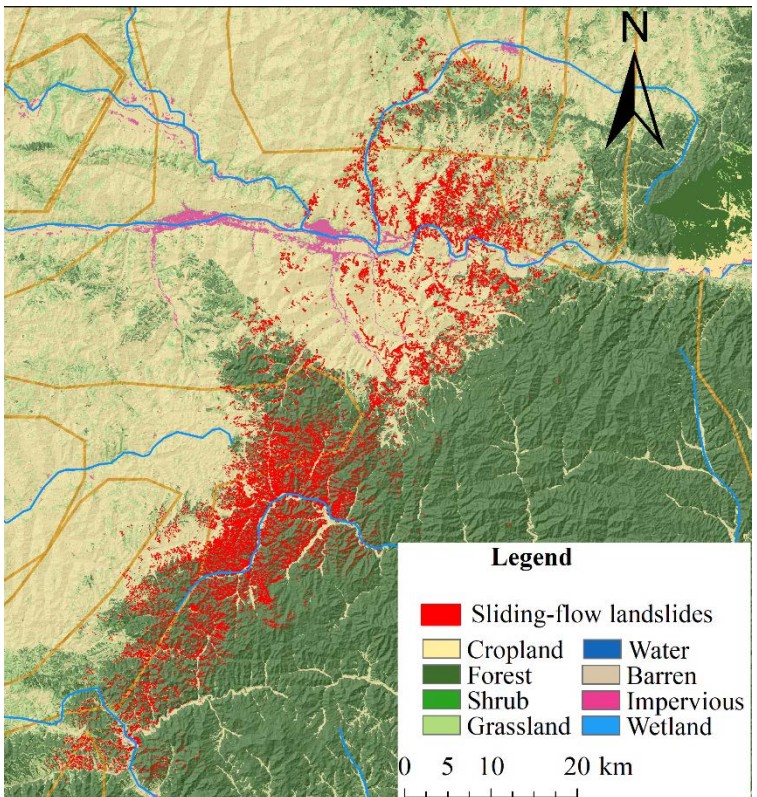

**Figure 17.** The "7.25" Tianshui sliding-flow landslides and land use in the study area (Base DEM data from https://geocloud.cgs.gov.cn/#/home, land use data from http://www.geodata.cn)

### 5.3 Sensitivity analysis

Dry loess has high cohesive strength but loses strength significantly when wetted (Derbyshire et al. 2001; Zhuang et al., 2018) and has water-sensitivity characteristics, with the strength parameters changing rapidly with the water content increasing. According to existing research, the cohesion of loess can be reduced from greater than 50 kPa at a low water content to less than 10 kPa at high water content (Zhuang et al., 2018). The internal friction angle of loess varies slightly, generally from about 25° in low water content to about 16° in a saturated state. To assess the loess strength influence on shallow loess landslides caused by prolonged heavy precipitation, the response of the slope stability to the strength change of saturated loess was calculated. Our results demonstrate that slope stability is greatly affected by cohesion, while the stability of the slope is less responsive to changes in the internal friction angle. Meanwhile, by changing the slope and fixing loess strength, the relationship between the safety factor and slope was obtained. The safety factor is variating obviously with slope changing, indicating that the cohesion and slope are key factors affecting soil stability and shallow landslides. Whereas the internal friction angle has minimal effect on shallow landslides in loess areas (Fig. 18).

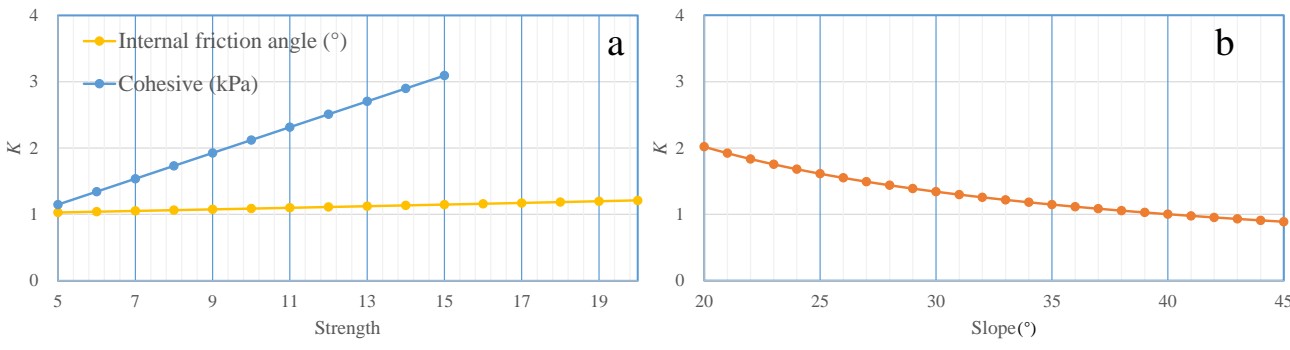

**Figure 18.** The safety factor varies with internal friction angle, cohesion changing, and slope.

## Conclusions

A rainfall-induced slope failure is a common cause of shallow landslides in the CLP. This study examines the shallow loess landslide triggered by prolonged heavy rainfall on July 25, 2013, in Tianshui, China, as a case study. The following results were obtained:

1. The "7.25" Tianshui sliding-flow landslide event triggered 47,005 landslides with a total area of 65.69 km² and the mean landslide area was 0.0013 km², which is smaller than most landslides triggered by other rainfall or earthquakes. Most of the landslides evaluated (80%) are smaller than 2,000 m² and landslides larger than 5,000 m² accounted for only 3%, indicating that the "7.25" Tianshui sliding-flow landslide events are primarily small landslides with group occurrences characters.

2. The $H/L$ ratio frequency ratio of the "7.25" Tianshui sliding-flow landslide event, ranged from 0.01 to 0.88 with a mean of 0.32. The equivalent coefficient of friction was below 0.17 for 16.85% of the loess landslides, indicating that the loess landslide travels via flow motion, resulting in sliding longer distances.

3. The Revised Infinite Slope Model (RISM) was proposed using an equal differential unit method that corrects for any deficiencies associated with increased safety factors on slopes that are invariable or increasing when the slope is greater than 40° according to the Taylor slope infinite model.

(4) The critical approximate liquid limit water content depth (also the sliding surface depth) of the shallow loess landslides with different slopes can be described as $h_w = 34.13a^{-1.15}$. The critical $I$-$D$ curve for slope instability for different slopes was constructed using the infiltration characteristics of loess in the study area.

*Code and data availability.* The data in this study were analysed with the Excel, and the figures were created with ArcViewTM GIS and Excel. All codes and date used in this work are available upon request.

*Author contributions.* Conceptualization, J. Z. and J. B.; methodology, J. Z. and C. H.; investigation, J. Z., J. B., J. X, and Y. Z.; data curation, J. Z. and J. X.; writing—original draft preparation, J. Z.; writing—review and editing, J. Z.; project administration, J. Z..; funding acquisition, J. Z. All authors have read and agreed to the published version of the manuscript.

*Competing interests.* The authors declare no conflict of interest.

*Acknowledgements.* The authors are very grateful to the anonymous reviewers and editors for their thoughtful review comments and suggestions which have significantly improved this paper. This study was financially supported by the National Natural Science Foundation of China: 42090053, 41922054. The authors thank AiMi Academic Services
(www.aimieditor.com) for the English language editing and review services.

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
