# Peer review of "Shallow landslides stability evaluation in loess areas according to Revised Infinite Slope Model: A case study of the 2013 "7.25" Tianshui sliding-flow landslide event in southwest of Loess Plateau, China"

_Natural Hazards and Earth System Sciences, 2022_

## Author Comment (AC1)

Dear Editors and Reviewer:

Thank you for your letter and for the reviewers' comments concerning our manuscript entitled "Characteristics and RISM of sliding flow landslides triggered by prolonged heavy rainfall in the loess area of Tianshui, China" (ID: nhess-2022-135). Those comments are all valuable and very helpful for revising and improving our paper, as well as the important guiding significance to our researches. We have studied comments carefully and have made correction which we hope meet with approval. The authors upload the both file (one is revised with tracking word and the other is revised without tracking word). The main corrections in the paper and the responds to the reviewer's comments are as flowing:

Responds to the reviewer's comments:

**Reviewer: 1**

1) the manuscript title is not suitable, and the manuscript title should contain the key words: prediction model etc. e.g., the title can be revised to: A novel prediction method of shallow landslide in loess area based on RISM, a "7.25" loess sliding-flow landslide event in 2013 in Tianshui, China case study.

Responded: Thank you for the suggestion, Considering the Reviewer's suggestion, the authors have changed the title to A novel prediction method of shallow landslide in loess area based on RISM, a "7.25" loess sliding-flow landslide event in 2013 in Tianshui, China case study.

2) Literatures seems to be extensive. Some recent research on shallow landslides, are suggested to be incorporated and supplemented. For example:
Fatma Keles and Hakan A. Nefeslioglu. Infinite slope stability model and steady-state hydrology-based shallow landslide susceptibility evaluations: The Guneysu catchment area (Rize, Turkey), CATENA, 2021.
Medina V et al. Fast physically-based model for rainfall-induced landslide susceptibility assessment at regional scale. CATENA, 2021.
Responded: Thank you for the suggestion, Considering the Reviewer's suggestion, the authors have added the recent research on shallow landslides in the section introduction.

3) the shallow loess landslide event name is not the same, e.g., "7.25 loess landslides" is used in the abstract, and "7.25" loess sliding-flow landslide events is used in text, please make it unified.
Responded: Thank you for the suggestion, Considering the Reviewer's suggestion, the authors have revised the shallow loess landslide event name through the text.

4) Line 85, In Section Introduction, please focus on scientific issues clearly and work to solve these problems well. This section needs definitely gives the research gap and objective about your work.

Responded: Thank you for the suggestion, Considering the Reviewer's suggestion, the authors have revised introduction and focus on scientific issues clearly and work to solve these problems well.

5) Do the earthquakes and symbols in Figure 1 represent the earthquake magnitude? If so, please add their units "Ms".
Responded: Thank you for the suggestion, Considering the Reviewer's suggestion, the authors have re-drawn the Figure 1.

6) English requires proofread by native speakers.
Responded: Thank you for the suggestion, the authors have polished the manuscript by native English speaker.

Yours sincerely,
Jianqi Zhuang etc.
E-mail: jqzhuang@chd.edu.cn

---

## Author Comment (AC2)

Dear Editors and Reviewer:

Thank you for your letter and for the reviewers' comments concerning our manuscript entitled "Characteristics and RISM of sliding flow landslides triggered by prolonged heavy rainfall in the loess area of Tianshui, China" (ID: nhess-2022-135). Those comments are all valuable and very helpful for revising and improving our paper, as well as the important guiding significance to our researches. We have studied comments carefully and have made correction which we hope meet with approval. The authors upload the both file (one is revised with tracking word and the other is revised without tracking word). The main corrections in the paper and the responds to the reviewer's comments are as flowing:

Responds to the reviewer's comments:

**Reviewer: 1**

The present study aims to improve prediction model for the shallow loess landslides induced by prolonged heavy rainfall.
The outstanding work is the improvement of the Taylor slope infinite model, based on equal differential unit method, permitting to correct the error when the safety factor increases with the increased slope when the slope is larger than 50°. Moreover, the intensity duration (I-D) prediction curve is proposed for the shallow loess landslides, considering the characteristics of rainfall infiltration, and different slopes.
Responded: Thank you very much for your comment. Just like the reviewer suggestion, the main purpose of this manuscript is to improve the deficiency that the safety factor invariable or increases with the slope increasing when the slope is larger than 40° calculated using the Taylor slope infinite model using equal differential unit method, and try to build a forecast model for rainfall-induced shallow landslides based on physical processes. The authors just improve the Taylor slope infinite model using equal differential unit method, and based on rainfall infiltration, a physical prediction model of rainfall-induced shallow landslides is proposed. The process proposed of prediction model of rainfall-induced shallow landslides in this study can provide an idea for subsequent software development and improved, and may be applied to many software, such as the SINMAP, SHALAD etc. Although there are some shortcomings, the author believes that the method proposed in this manuscript has a good reference for future shallow landslide prediction and promote the integration of statistical and physical models.

But several misunderstanding are encountered in this paper:
- the study is focused on the development of RISM model, for a better estimation of safety factor for slope higher than 50°. However, as seen from field data (in figure 7), all landslides are triggered in slopes lower than 35°. In that way, we do not really understand why the development of this model can help in quantifying the stability of slope in this context.
Responded: Thank you very much. This is a particularly good suggestion. It is difficult to reach the landslide location when the slope greater than 35° and conduct field measurements during the field investigations. So, the 89 landslides measured in the field investigation were within the range of 35°. However, it can be seen from the figure 1 that the "7.25" Tianshui sliding-flow landslides is mainly distribution between 20-50°, and more than 50% of the sliding-flow landslides are located in slope with greater than 35° according to all the "7.25" Tianshui sliding-flow landslides data. the main purpose of

this manuscript is to improve the deficiency that the safety factor invariable or increases with the slope increasing when the slope is larger than 40° calculated using the Taylor slope infinite model using equal differential unit method, and try to build a forecast model for rainfall-induced shallow landslides based on physical processes. The I-D prediction model based on physical process proposed in this manuscript has a clear physical process considered different slope characteristics can give the higher prediction accuracy.

[Figure]

Fig. 1 The distribution of the "7.25" Tianshui sliding-flow landslides vs slope

- Moreover, the validation of the approach (notably the I-D curve for different slopes) is not realised, as the I-D curves for different slopes are not tested in real case for all landslides.
Responded: Just like the reviewer comment, the authors did not calculate the I-D curve for all slopes. The "7.25" Tianshui sliding-flow landslides is mainly distribution between 20-50°, and more than 50% of the sliding-flow landslides are located in slope with greater than 30-45° according to all the "7.25" Tianshui sliding-flow landslides data. So, I-D curves for different slopes in the manuscript are not tested suing the range of 30-45 degrees (Figure 14). Calculating all the slope I-D curves has little effect on the regional shallow landslide prediction. It should calculate the I-D curve according to the dominant slope or the dominant slope of shallow landslides for regional shallow landslide prediction. And, the I-D curve of a single slope can be built according to the slope gradient and physical parameters for the prediction of a single slope.

- Several key information and references are missing, such as: the methodology for obtaining the landslide's map; the justification about how the 89 studied landslides are representative of the 45 000 landslides; the initial  I-D curve that has been used in this study.
Responded: Thank you very much. The authors have added the information about the map for obtaining the landslide's map in section 3.1. Now, many high-resolution satellite data are free, such as GE and other data. The author uses the google earthed data source, which provides historical images, which can be used for sliding-flow landslide interpreted. Remote sensing images (~2 m resolution from Google earth images) from October 2012 (before the sliding-flow landslide event) and December 2013 (after the sliding-flow landslide event), before and after the sliding-flow landslides were used for sliding-flow

landslide interpreted in this manuscript. For the question of how 89 landslides represent 45,000 landslides, it is impossible to obtain the sliding surface depths of 45,000 landslides, because measuring the sliding surface depths requires manual measurement in the field investigation. The authors mainly want to give the information that the relationship between the sliding surface depth and slope, area of the landslides using the 89 landslides. Therefore, the relationship between the sliding surface depth and the slope, area of the 89 landslides in this manuscript is to explain the close relationship between the sliding surface depth and the slope. It provides data support for constructing I-D curves under different slopes.

Based on these main elements, I suggest to reject the manuscript.

You can also find below some additional comments:

Line 89 "7.25" loess sliding-flow landslide events in Tianshui Gansu province : please define the meaning of "7.25", otherwise you have to add the reference to this.

Responds: It is very good suggestion, thanks very much. Considering the Reviewer's suggestion, the authors have changed the landslide event to 2013 Tianshui sliding-flow landslide event and added information about the meaning of 2013 Tianshui sliding-flow landslide event according to the landslide occurred date in the section 1. (???).

Figure 1 is not a geomorphological nor geological map; please put elements concerning these 2 features ; moreover, earthquake location is not usefull to this paper, as the study works on landslides induced by heavy precipitations, and not by earthquake.

Responded: Thank you for the reviewer's suggestion. Considering the Reviewer's suggestion, the authors have redrawn the Figure 1 and added the topographic, land use and loess thickness that have the most direct impact on shallow landslides occurrence in the Figure 1.

I suggest to dedicate paragraph 2 to geological, geomorphological, and climate characteristics ; as the threshold of these landslides is heavy precipitation, I recommend to detail the climate description in a sub-paragraph 2.2 (subparagraph 2.1 could be geological and geomorphological characteristics) ; finally I suggest to move 2.2 to paragraph 3, as it concerns landslides features.

Responded: Thank you for the reviewer's suggestion. Considering the Reviewer's suggestion, the authors have revised section 2 in detail. In section 2.1, the authors focused on describing the geographic, geomorphological, and climate characteristics of the region; in section 2.2, the authors introduced the climate of the region, especially the information on extreme rainfall events in recent decades. And, transfer the landslide information to section 3.

I think it is necessary to detail input data and the processes to obtain landslides maps, with some zoom on figure 2.

Responded: Thank you very much. Considering the Reviewer's suggestion, the authors have added the information about the map for obtaining the landslide's map in section 3.1. Now, many high-resolution satellite data are free, such as GE and other data. The author uses the google earthed data source, which provides historical images, which can be used for sliding-flow landslide interpreted. Remote sensing images (~2 m resolution from Google earth images) from October 2012 (before the sliding-flow landslide event) and December 2013 (after the sliding-flow landslide event), before and after the sliding-flow landslides were used for sliding-flow landslide interpreted in this manuscript. By comparing the remote sensing images before and after the event, the shallow landslide induced by the 2013 Tianshui sliding-flow landslide event can be clearly distinguished, as shown in the flowing figure.

[Figure]

Figure 2 Remote sensing images before and after the event. (a: before the sliding-flow landslide event, b: after the sliding-flow landslide event)

table 2 : the term "area" is not appropriate ; you can use "surface area"

Responded: Thank you very much. Considering the Reviewer's suggestion, the authors have used "surface area" replace the term "area" in the table 2.

Line181 : you have to indicate and discuss whether these 89 landslides are statistically representative of the 47 005 landslides ; you also have to precise the characteristics of the landslide you consider for obtaining the depth of the landslide.

Responded: Thank you very much. For the question of how 89 landslides represent 45,000 landslides, it is impossible to obtain the sliding surface depths of 45,000 landslides, because measuring the sliding surface depths requires manual measurement in the field investigation. The authors mainly want to give the information that the relationship between the sliding surface depth and slope, area of the landslides using the 89 landslides. Therefore, the relationship between the sliding surface depth and the slope, area of the 89 landslides in this manuscript is to explain the close relationship between the sliding surface depth and the slope. It provides data support for constructing I-D curves under different slopes.

The authors very agree with the comment that precise the characteristics of the landslide you consider for obtaining the depth of the landslide. It is difficult to reach the landslide location when the slope greater than 35° and conduct field measurements during the field investigations. So, the 89 landslides measured in the field investigation were within the range of 35°. However, the "7.25" Tianshui sliding-flow landslides is mainly distribution between 20-50°, and more than 50% of the sliding-flow landslides are located at slope with greater than 35° according to all the "7.25" Tianshui sliding-flow landslides data. The authors mainly want to give the information that the relationship between the sliding surface depth and slope, area of the landslides using the 89 landslides. Therefore, the relationship between the sliding surface depth and the slope, area of the 89 landslides in this manuscript is to explain the close relationship between the sliding surface depth and the slope. It provides data support for constructing I-D curves under different slopes.

figure 7 is confusing, as slope and surface areas are in the same graph. 2 separate graphs might be better

Responded: Thank you very much. Considering the Reviewer's suggestion, the authors have separated the graphs in the figure 7 and made it clearer.

line 200 : it is necessary to provide details and possible explanation about "a certain depth of loess becoming close to liquid limit water content" : which depth? Depending on what?

Responded: Thank you for the suggestion. In the section, the authors described the mode and process of sliding-flow landslide induced by prolonged heavy rainfall. According to a large number of field monitoring and previous research results, this specific depth is generally considered to be a depth of 1-2m, the author adds references in the sentence.

line 209 : please provide references concerning the depth of shallow landslides in loess.
Responded: Thank you for the suggestion, the authors have added references in the sentence.

P12 line 214 : it is mentioned that "loess landslide transformation to mudflows occur most often on slopes of 25 – 45°" ; why is it different from the results shown on figure 7? Please explain.
Responded: Thank you very much. It is difficult to reach the landslide location when the slope greater than 35° and conduct field measurements during the field investigations. So, the 89 landslides measured in the field investigation were within the range of 35°. However, it can be seen from the figure 1 that the "7.25" Tianshui sliding-flow landslides is mainly distribution between 20-50°, and more than 50% of the sliding-flow landslides are located in slope with greater than 35° according to all the "7.25" Tianshui sliding-flow landslides data.

line 220 ; provide reference of Taylor;
Responded: Thank you for the suggestion, the authors have added references in the sentence.

line 222 ; the length of the body is b, not l
Responded: Thank you for the suggestion, the authors have revised the sentence and changed the *l* to *b*.

figure 9 ; I don't understand the different directions of arrows designing the flow lines ?
Responded: Thank you for the suggestion, the figure 9 is drawn with reference to Tylor's infinite slope model, the authors have added references.

Fig 10 : how do you chose the value c' phi', gsat ? for high slopes, I am not sure that such value can be found in the field
Responded: Thank you very much. The Figure 10 is to illustrate the phenomenon that the safety factor invariable or increases with the slope increasing when the slope is larger than 40° calculated using the Taylor slope infinite model using equal differential unit method. So the value of the c' phi', gsat is used the mean strength of saturated loess in the loess area which obtained based on loess at 25% water content (saturation 84%). The figure just shows that Taylor slope infinite model cannot be used to calculate the safety factor of shallow landslides when the slope is higher than 4°. The authors have added the information in the text.

p15, line 292 : please provide some details or references on loess test results ; the figure 13 doesn't provide these elements
Responded: Thank you for the suggestion, the authors have added the test method and the value of the c,and phi' used in the Figure 13 has provide in the text. The value of the c,and phi' used in the Figure 13 is based on the mean strength of saturated loess in the study area.

line 298 : it is necessary to plot this law on figure 13
Responded: Thank you for the suggestion, the authors have added the law on figure 13.

figure 14 : we don't know what is Yan'an "7.13" group shallow landslide . Which are the values of the characteristics used here (C, phi, rainfall events, slopes ….)? more generally, how are constructed these curves ? is it still the parameters c, phi, … from figure 10 that are considered?

Responded: It is very good suggestion, thanks very much. The value of the c,and phi' used in the Figure 14 iare same as the Figure 13 which has provide in the text. The value of the c,and phi' used in the Figure 13 is based on the mean strength of saturated loess in the study area.  Considering the Reviewer's suggestion, the author have added the information in the Figure 13 and Figure 14. The Yan'an "7.13" group shallow landslide was also a group-occurring shallow sliding-flow landslide event induced by prolonged heavy rainfall in 2013. The loess in the Yan'an and Tianshui have similar properties and both belong to the Loess Plateau. Therefore, the authors used the both mass shallow sliding-flow landslide events in 2013 to verify the feasibility of the I-D curve built in the manuscript. The I-D curve of Figure 14 is based on Equations 8 and 9, where hw is from Figure 13. According to different slopes, referring to the parameters in Figure 10, the Figure 14 can be obtained. The authors have added information in the text.

I don't see here really validation of the model or approach ; you could apply this I-D curve on the all area of study, with considering for each local area or pixel I-D curve, considering local parameters. I don't understand how you can consider the "7.25 loess sliding-flow events" as a unique point on your analysis.

Responded: Thank you for the suggestion. The I-d curve is the most common and widely used method for predicting regional shallow landslides currently. The authors give the I-D curves under different slopes in Figure 14 and uses 7.25 Tianshui sliding-flow landslide event for verification. For the question of why a regional landslide event has only one point, because rainfall events are regional, a rainfall event has only one rainfall duration and intensity, and the induced slopes may be located on different slopes. Calculating all the slope I-D curves has little effect on the regional shallow landslide prediction. It should calculate the I-D curve according to the dominant slope or the dominant slope of shallow landslides for regional shallow landslide prediction. And, the I-D curve of a single slope can be built according to the slope gradient and physical parameters for the prediction of a single slope. At present, there are only a few mass sliding-flow landslide events in loess area, and there are only these two events with detailed i-d records, so the author uses these two events for verification.

figure 15 : legends of the curves are missing, we have no information on them

Responded: Thank you for the suggestion, the authors have added the legend in the Figure 15.

paragraph 5.2 : why this part is within the discussion paragraph ?

Responded: Considering the Reviewer's suggestion, the authors have deleted the section of sensitivity analysis and added a section to discuss the limitations of the model and discuss the impact of vegetation on sliding-flow landslides due to many sliding-flow landslides occur in areas with better vegetation coverage.

As a general comment, the discussion is not really realised.

Responded: Considering the Reviewer's suggestion, the author has added a section to discuss the limitations of the model and discuss the impact of vegetation on sliding-flow landslide, because many sliding-flow landslides occur in areas with better vegetation coverage.

Figure 16 : as figure 7, the figure is confusing ; it is better to provide One graph for each parameter;

Responded: Thank you for the suggestion, the authors have deleted the Figure 16.

line 349 : there are errors in internal friction angle, expressed in kPa!
Responded: Thank you for the suggestion, the authors have revied in the text.

The paper requires proofread by English native speakers, as it encounters several English mistakes or inadequate words, as well as sentences wrong formulation. Indeed, several sentences are not clear; for instance:
1, line 13
1, lines 16 to 18 : sentence too long
3, line 95 : you mean "the elevation of the study area" ? please check
17, line 331 : non comprehensive sentence
17, line 333, 334 : I don't understand
Responded: Thank you for the suggestion, the authors have polished the manuscript by native English speaker.

Yours sincerely,
Jianqi Zhuang etc.
E-mail: jqzhuang@chd.edu.cn

---

## Author Response (AR2)

Dear Editors and Reviewer:

Thank you for your letter and for the reviewers' comments concerning our manuscript entitled "A novel prediction method for shallow landslides in loess areas: A case study of the 2013 "7.25" Tianshui sliding-flow landslide in Gansu province, China" (ID: nhess-2022-135). Those comments are all valuable and very helpful for revising and improving our paper, as well as the important guiding significance to our researches. We have studied comments carefully and have made correction which we hope meet with approval. The authors upload the both file (one is revised with tracking word and the other is revised without tracking word). The main corrections in the paper and the responds to the reviewer's comments are as flowing:

Responds to the reviewer's comments:

**Editor comments:**

in view of the two reviewers' reports and after careful consideration of the manuscript and the issues raised by referee #3 after the major revision iteration, we still fall under the situation of the first review round, with one review which is good (minor revision) while the other recommends rejection. Due to this problem, I am forced to ask again for major revisions of the manuscript, as most of the flaws that were present in the original version seem to be still present.

My suggestion is to carefully read and try to address the comments of review #3, in particular:

Responds: Yes, we have read the comments of the both reviewers, and indeed their comments differ significantly. The main issue is the applicability of the Revised Infinite Slope Model (RISM). As we know, modifying or establishing a new model, or extending the application of a model to other areas, requires multiple revisions and improvements. The model we have established is no exception. The objective of the manuscript is proposing the Revised Infinite Slope Model (RISM) that corrects for any deficiencies associated with increased safety factors on slopes that are invariable and predicting regional shallow landslide. However, we cannot deny that there are shortcomings in this revised model, such as limited applicability to loose layer thickness and uneven rainfall infiltration. These situations are not adaptable and require further revision.

- please clarify the very important issue raised by the reviewer on the absence of soil on steep slopes of the study area, a thing which evidently would hinder the possibility of the development of shallow slides

Responds: It is very good suggestion, thanks very much. The aim of this manuscript is to study soil landslides. The infinite slope model is also suit for the soil landslide. This manuscript is based on the revised and improved form infinite slope model, so the Revised Infinite Slope Model (RISM) is only applicable to shallow soil landslides.

- also, if the slides are not shallow and affect the bedrock, the model you use (infinite slope) is probably not well suited to represent the geotechnics of the slope and therefore not apt at computing a reliable factor of safety

Responds: This issue is the same as the previous, because the infinite slope model has two hypotheses: one is that the length of the slope is much greater than the depth, and the second is the slope is mean soil slope. The manuscript is to study the loess (soil) shallow landslides and the

manuscript is based on the revised and improved form infinite slope model, so the Revised Infinite Slope Model (RISM) is only applicable to shallow soil landslides.

- please also explain and justify why those hypothetical shallow (or soil) slides would be triggered by long-duration rainfall, something that is contrary to prior knowledge

Responds: The manuscript is focused on the prediction of rainfall-induced loess shallow landslides. Additionally, the infinite slope model is specifically designed for forecasting rainfall-induced shallow landslides as well. Considering the research findings on the triggering factors of loess shallow landslides, it is evident that loess landslide occurrences are most sensitive to long-duration rainfall. Only prolonged rainfall can lead to soil saturation and the formation of sliding-flow characteristics. Therefore, the manuscript is the prediction of rainfall-induced loess shallow landslides.

- another important point: if the geotechnical properties of the affected materials are as stated (cohesion 5 kPa and friction 20°), how can you justify the stability of slopes with gradients overcoming 50°? This simple fact strongly compromises the basis of your work, according to reviewer #3 and according to the old report of reviewer #2, whose comments have not been answered properly. The response to this important point cannot be verbal but should include new data and maps demonstrating the validity of the basic hypotheses cited above as well as documenting the spatial distribution of slope angle and past landslide occurrences over the area, with some accompanying spatial statistics

Responds: It is very good suggestion, thanks very much. This is one of the contents to be studied in this manuscript. Due to the issue of the increase of safety factor in the infinite slope model with increasing slope (>45°) in calculating the stability of shallow landslides, this manuscript proposes the Revised Infinite Slope Model (RISM) using an equal differential unit method to correct for deficiencies when the safety factor remains unchanged or increases with increasing slope, greater than 40° as calculated using the Taylor slope infinite model. The slopes in the Loess Plateau region often have slopes above 45°, and even up to 80° (the slope is more than 40° account for 5.3%). At the same time, the slopes where shallow landslides occur in the Loess Plateau region often exceed 40° (the landslide is more than 50° account for 13.81%). Therefore, many slope stability calculation models and even soil erosion models are not applicable to the Loess Plateau region. The Revised Infinite Slope Model (RISM) proposed in this manuscript, in calculating slope stability, found that with increasing slope, the critical depth of the soil layer induced by rainfall-induced slope landslides decrease. So, at slopes greater than 50°, many shallow landslides often have a depth of only 10-20cm, which in the Loess Plateau region are often considered as soil erosion or surface soil movement. Therefore, while correcting the infinite slope model, this manuscript also provides the critical depth at different slopes and combines rainfall infiltration to calculate the critical rainfall forecast model for shallow landslides at different slopes in CLP. At the same time, this manuscript focuses only on rainfall-induced shallow landslides and restricts the application of the model to the Loess Plateau region. This model cannot be applied to landslides that are controlled by the strength of the soil structure or the structure and fissures of the rock-soil mass.

[Figure]

Figure 1 The slope distribution of the landslide and landscape

- finally, the infinite slope model makes the assumption that the sliding depth decreases with increasing slope gradient but the model doesn't take into account that also soil depth decreases as well with slope angle. According to reviewer #3, this seriously hampers the applicability of the method. This point as well needs an in-depth explanation and some counterproof to be considered valid.

Responds: It is very good suggestion, thanks very much. The study area in this manuscript is the Loess Plateau, and at the same time, this manuscript only focuses on shallow sliding-flow landslides induced by rainfall, and the application area of the model is limited to homogeneous soil cover areas. This model cannot be applied to landslides that are subject to the strength of the soil structure or the sliding controlled by the rock and soil structure or cracks.

In the Loess Plateau region, the thickness of loess often reaches more than 100m, and this thickness does not change with the change of slope, so this paper does not consider the issue that soil thickness decreases with the increase of slope.

**Reviewing: 1**

The authors mentioned that the main purpose of the manuscript is to improve the deficiency that the safety factor invariable or increases with the slope increasing when the slope is greater than 50° when Taylor infinite slope model is adopted. Then, to show that large amount of landslides was occurred in steep slopes (greater than 50°), the authors provide the Figure 1 in the "author response". But I could not understand how the shallow landslides has occurred in steep slope whose slope angle is greater than 50°. This is because shallow landslide could be occurred natural slope whose depth is less than 1-2m and steep natural slope (> 50°) does have this soil cover over bedrock.

Responds: Thank you very much. The main purpose of this manuscript is to revise the infinite slope model so that it can be applicable to steep slopes in the Loess Plateau region. The Loess Plateau region is widest distribution of soil slopes in China, where the thickness of loess often exceeds 100 meters. Loess exhibits typical water-sensitive characteristics and is prone to catastrophic failure and the formation of sliding-flow landslides. For example, the prolonged heavy rainfall in 2013, 2015, and 2021 resulted in thousands of sliding-flow landslide events in the Loess Plateau region.

Based on the distribution of these landslide events and field investigations, it is observed that many shallow landslides in the Loess Plateau region occur on slopes with gradients of over 40 degrees

(~10-15%, Zhuang et al., 2017; Shao et al., 2023). This is because the accumulation thickness of loess in the Loess Plateau region is large, and it has little relationship with the slope gradient. As a result, the risk and probability of sliding-flow landslides still exist on steep slopes (50°) in the Loess Plateau region.

At the same time, this manuscript focuses only on rainfall-induced shallow landslides and restricts the application of the model to the Loess Plateau region. This model cannot be applied to landslides that are controlled by the strength of the soil structure or the structure and fissures of the rock-soil mass.

[Figure]

Figure 2 The slope distribution of the landslide and landscape (a, in this study; b, from Shao et al., 2023)

The authors considered that the shallow landslides were occurred in this study area due to prolong heavy rainfall and adopted infinite slope model in this analysis. The shallow landslide is typically occurred in natural slopes whose slope depth is less than 1 – 2 m. However, as shown figure 7(b), the steep slope (> 50°) will have very thin or almost 0 m soil depth. Therefore, the occurrence of shallow landslide is not practically possible. Or the landslide occurred in slopes with greater 50° may not be the shallow landslide, which means that infinite slope model cannot apply to those steep slope.

Responds: Thank you very much. As mentioned earlier, The Loess Plateau region is widest distribution of soil slopes in China, where the thickness of loess often exceeds 100 meters. Loess exhibits typical water-sensitive characteristics and is prone to catastrophic failure and the formation of sliding-flow landslides. For example, the prolonged heavy rainfall in 2013, 2015, and 2021 resulted in thousands of sliding-flow landslide events in the Loess Plateau region.

In this study, the revised infinite slope model was used to calculate slope stability. It was found that with increasing slope, the critical approximate liquid limit water content layer gradually decreases, but the rate of decrease slows, from 1.14 m at 20° to 0.47 m at 40°. This relationship can be expressed via the power-law. Therefore, when the slope is greater than 50 degrees, many shallow landslides have a sliding thickness of only 10-20cm, which is often attributed to soil erosion or surface soil movement in the Loess Plateau. Meanwhile, this study also provided the critical depth for different slopes and calculated the critical rainfall forecast model for shallow landslides under different slopes, considering rainfall infiltration. The steeper slopes have a higher probability of shallow landslides occurring, as rainfall infiltrates the soil and reduces its strength at a certain depth, triggering soil instability. It should be noted that this study focuses only on rainfall-induced shallow landslides and flow, and the application of the model is limited to the loess region. It cannot be

applied to landslides controlled by soil structure strength, or rock-soil structures, and fractures.

In addition, cohesion and friction angle of slope materials in the study are 5 kPa and 20 °, respectively. Therefore, when we consider the cohesion and friction angle values (somewhat lower than typical values in other area), it is not easy to understand that this soil material can have steep slope over 50°.

Responds: Yes, it is certain that the loess will failure when the slope is greater than 50° with such low soil strength, However, the focus of this manuscript is to discuss the applicability of the infinite slope model under different slope, and to determine the relationship between the critical depth and slope required to trigger loess sliding-flow under rainfall. It has been found that with increasing slope, the critical approximate liquid limit water content layer gradually decreases, but the rate of decrease slows, from 1.14 m at 20° to 0.47 m at 40° according to field investigations and calculate. This relationship can be expressed via the power-law. This why we often observe shallow landslides occurring in steep areas with relatively low rainfall intensity. Therefore, when the slope is greater than 50 degrees, many shallow landslides have a sliding thickness of only 10-20cm, which is often attributed to soil erosion or surface soil movement in the Loess Plateau.

In order to persuade readers, the authors should provide the slope angle distribution map of the study area. In addition, the authors should provide the any pictures or proof that showing the shallow landslide occurred in high slope angle.

Responds: Thank you very much. The slopes in the Loess Plateau region often have slopes above 45°, and even up to 80° (the slope is more than 40° account for 5.3%). Based on the distribution of these landslide events and field investigations, it is observed that many shallow landslides in the Loess Plateau region occur on slopes with gradients of over 40 degrees (~10-15%, Zhuang et al., 2017; Shao et al., 2023). The maximum slope observed for shallow landslides is 69°, while the maximum slope of the study area's topography is 71° (Shao et al., 2023).

[Figure]

Figure 3 The slope distribution of the landslide and landscape (a, in this study; b, from Shao et al., 2023)

In addition, I should point out a serious problem in the authors' basic idea. As the authors showed in Fig. 7(b), the sliding depth decreases when slope angle increases. When slope is greater 50°, the we can imagine that the slope depth could be very shallow or almost close to 0 m. However, the authors did not consider that the slope depth could be decreased to very shallow or almost 0m when slope angle increases. As shown in Fig. 10, the authors considered fixed slope depth even if slope

angle increases over 50° and I believe it causes miscalculation. Therefore, the revised infinite slope model could not be needed.

Responds: It is very good suggestion, thanks very much. Figure 7 presents the depth distribution of landslides induced by prolonged heavy rainfall based on field investigation. As mentioned in our manuscript, we surveyed approximately 83 landslide samples and obtained slope and sliding depth distribution maps. We found that the depth of the sliding-flow landslide had no correlation with the landslide area and had a negative correlation with the slope. With increasing slope, the depth of the landslide decreases, that is, the greater the slope, the shallower the sliding surface, and the smaller the slope, the greater the depth of the sliding surface. These findings align with the real observations we made in the field. However, as the slope continues to increase, the depth of the sliding surface decreases. Based on previous experience, sliding surfaces with depths within 10 cm are also common occurrences, especially in the Loess Plateau region. When rainfall infiltrates into the loess, the strength of the loess decreases. Shallow sliding-flow can be triggered in steep slope once the infiltration reduces the strength to a certain extent. This is why we often see bare areas in steeper slope regions of the Loess Plateau after rainfall because these steeper slope slopes are prone to shallow sliding-flow during prolonged rainfall. Through our statistics on existing landslides, we found that shallow sliding-flows with slopes greater than 50° accounted for 13.81% of the landslides induced by prolonged heavy rainfall in Tianshui in 2013. Furthermore, we compared the slope distribution of the landslides that occurred in this event with other interpreted distributions and found that landslides with slopes greater than 50° accounted for a certain proportion of the shallow sliding-flows induced by continuous heavy rainfall in Tianshui in 2013. The maximum slope observed for shallow landslides was 69°, while the maximum slope of the study area's topography was 71° (Shao et al., 2023).

[Figure]

Figure 4 Map showing the relationship between the LAD and slope factors (from Shao et al., 2023)
Table 1 Statistical slope factor for the landslides and landscape area (from Shao et al., 2023)

| Variable | | Landslide | Landscape |
|---|---|---|---|
| Hillslope gradient [°] | Mean | 25.5 | 20.5 |
| | Max | 69 | 71 |
| | Min | 0 | 0 |

**Reviewing: 2**

The manuscript was thoroughly revised according to the comments of reviewers. I suggest it should be accepted after a minor revision.

1.I suggest the title should be modified as "Shallow landslides stability evaluation in loess areas according to Revised Infinite Slope Model: A case study of the 2013 "7.25" Tianshui sliding-flow landslide in Gansu province, China"

Responds: It is very good suggestion, thanks very much. We have revised the manuscript title according to this suggestion.

2.The technical terms should be consistent through the whole manuscript, i.g. "liquid limit water content" should also be used in abstract instead of " liquid limited water content". Such mistakes should be avoided.

Responds: Thank you very much. The authors have revised the technical terms which not consistent in the text.

3.The English in this manuscript should be further revised.

Responds: Thank you very much. The manuscript has been polished by native English speaker.

Shao, X.; Ma, S.; Xu, C.; Xu, Y. Insight into the Characteristics and Triggers of Loess Landslides during the 2013 Heavy Rainfall Event in the Tianshui Area, China. Remote Sens. 2023, 15, 4304. https://doi.org/10.3390/rs15174304

Zhuang, J.Q., Peng, J.B., Wang, G.H., Javed, I., Wang, Y. and Li, W.: Prediction of rainfall-induced shallow landslides in the Loess Plateau, Yan'an, China, using the TRIGRS model, Earth Surface Processes and Landforms, 42(6), 915-927, doi:10.1002/esp.4050, 2017

Jiqnzi Zhuang, Jianbing Peng and Chenhui Du
jqzhuang@chd.edu.cn

---

## Author Response (AR3)

Dear Editors and Reviewer:

Thank you for your letter and for the reviewers' comments concerning our manuscript entitled "A novel prediction method for shallow landslides in loess areas: A case study of the 2013 "7.25" Tianshui sliding-flow landslide in Gansu province, China" (ID: nhess-2022-135). Those comments are all valuable and very helpful for revising and improving our paper, as well as the important guiding significance to our researches. We have studied comments carefully and have made correction which we hope meet with approval. The authors upload the both file (one is revised with tracking word and the other is revised without tracking word). The main corrections in the paper and the responds to the reviewer's comments are as flowing:

Responds to the comments:

You replied to the criticism raised, and despite some limitations, the paper could now be ready for publication. However, I suggest adding 1 figure showing 2-3 photos of typical shallow landslides on steep slopes in the study area to better help the reader in understanding the process described. Responds: It is very good suggestion, thanks very much. The authors have revised the figure 2 and added 4 photos of typical shallow landslides on steep slopes in the study area to help the reader in understanding the process described. Please see the figure 2.

I suggest to further check the English form.
Responds: Thank you very much. The manuscript has been polished by native English speaker.

Others, the authors have revised the technical terms which not consistent in the text and added related references which recently published in the text.

Jiqnzi Zhuang, Jianbing Peng and Chenhui Du
jqzhuang@chd.edu.cn

---

## Author Response (AR4)

Dear Editors and Reviewer:

Thank you for your letter and for the reviewers' comments concerning our manuscript entitled "A novel prediction method for shallow landslides in loess areas: A case study of the 2013 "7.25" Tianshui sliding-flow landslide in Gansu province, China" (ID: nhess-2022-135). Those comments are all valuable and very helpful for revising and improving our paper, as well as the important guiding significance to our researches. We have studied comments carefully and have made correction which we hope meet with approval. The authors upload the both file (one is revised with tracking word and the other is revised without tracking word). The main corrections in the paper and the responds to the reviewer's comments are as flowing:

Responds to the comments:

Your paper is ready for publication, however, I suggest some technical improvements to Fig. 1, Fig. 2, and Fig. 17. They are not aligned with the high-quality-style standard of our journal and in general, of a Q1 journal: legend and scale bars need to better standardized and with a suitable proportional "view", the name "legend" is not necessary. I suggest to consider as comparative example figures that usually appear in our journal or Nature/Science journal.
Responds: It is very good suggestion, thanks very much. The authors have revised the figure 1, 2 and 17 to improve the figure quality standard according to nhess journal or Nature/Science journal. Please see the figure 1, 2 and 17.

Jiqnzi Zhuang, Jianbing Peng, Chenhui Du, Yi Zhu and Jiaxu Kong
jqzhuang@chd.edu.cn